# Multi-Objective Protein Design via Memory-Aware Test-Time Scaling in Diffusion Models

**Ming Yang** [1]  **Xin Zheng** [2]  **Yi Li** [3]  **Yizhen Zheng** [4]  **Huan Yee Koh** [5]  **Yanqing Guo** [3]  **Xiaofeng Cao** [6]  **Shirui Pan** [7]

## Abstract

Multi-objective protein design is essential for meeting the complex demands of synthetic biology. To adapt to shifting multi-functional targets without the prohibitive cost of retraining, test-time scaling has emerged as a flexible, training-free alternative. However, current test-time diffusion methods face critical challenges: i) *ineffective learning from interaction history leading to repetitive design errors,* ii) *over-reliance on successful cases as the reward signal,* and iii) *difficulties in balancing multi-objective functional trade-offs.* To address these limitations, we propose **MoMST**, a framework for **M**ulti-**o**bjective protein design via **M**emory-aware **S**elf-contrastive learning with **T**est-time scaling in diffusion models. At test time, we develop a memory bank to extract generalizable reasoning experience from historical iterations. Building on this powerful experience learner, we derive rich residue-level relative preference signals from both successful and failed cases via self-contrastive learning for guiding protein generation. To ensure balance among competing multi-objective functions, we present an inference-time Pareto alignment strategy to resolve objective conflicts. Evaluations on both single-objective and complex multi-objective tasks demonstrate the remarkable performance of MoMST, with code available at https://github.com/MingYangi/MoMST.

## 1. Introduction

Designing functional proteins for applications is crucial in synthetic biology and drug development (Jiang et al., 2025; Song et al., 2024; Sun et al., 2025). Traditional approaches primarily focus on optimizing a single function or property (Jin et al., 2025; Sun et al., 2025), yet protein applications in real-world scenarios necessitate simultaneously meeting multiple complex attributes. For instance, therapeutic protein design optimizes critical structural metrics, including stability, hydrophobicity, and surface accessibility as well as reducing immunogenicity risk and enhancing solubility (Rosenberg, 2006; Goldenzweig & Fleishman, 2018; Gainza et al., 2023). This requires a shift toward multi-objective protein design (Chen et al., 2025; Liu et al., 2024), aiming to identify optimal sequences under conflicting biophysical constraints to enhance the practicality of proteins.

Recently, diffusion-based generative models have emerged as a powerful paradigm for protein design, demonstrating a remarkable ability to capture the complex distribution of natural proteins (Winnifrith et al., 2024; Koh et al., 2025; Alamdari et al., 2023; Zhou et al., 2025). However, most existing methods focus predominantly on single-objective design. Even when extended to multi-functional scenarios, these models typically require expensive retraining for specific targets. Such a dependency hinders rapid adaptation to new requirements and limits the flexible balancing and combining of diverse design goals. To address this gap, recent studies have shifted toward **test-time scaling**, which leverages distinct design objectives as reward functions in inference to guide diffusion model denoising for target-compliant protein sequence generation (Uehara et al., 2025b). Typically, existing test-time reward optimization methods (Dhariwal & Nichol, 2021; Wu et al., 2023; Li et al., 2024) use a single-shot denoising pass for test-time generation, while others (Uehara et al., 2025a) refine sample quality via cyclic noising and reward-guided denoising with iterative refinement frameworks.

Despite the powerful performance of test-time scaling paradigm based on diffusion models, existing methods still face significant challenges, especially in multi-objective landscapes: (C1) Limited ability to balance multiple functional objectives: Most existing test-time scaling methods

---

[1]School of Information and Communication Engineering, Dalian University of Technology [2]School of Computing Technologies, RMIT University [3]School of Future Technology, Dalian University of Technology [4]Department of Cancer Medicine, Monash University [5]Department of Data Science and AI, Monash University [6]School of Computer Science and Technology, Tongji University [7]School of Information and Communication Technology, Griffith University. Correspondence to: Yanqing Guo <guoyq@dlut.edu.cn>.

*Proceedings of the 43$^{rd}$ International Conference on Machine Learning*, Seoul, South Korea. PMLR 306, 2026. Copyright 2026 by the author(s).

focus on single-objective protein design. Even though (Uehara et al., 2025a) has made an attempt at multi-objective design, it simply uses linear weighted summation, which introduces dominant-objective bias and fails to capture complex trade-offs under biological constraints, leading to significant limitations in balancing multiple functional objectives. (C2) Failure to learn from iterative sampling history: In the iterative sampling, existing methods generate new candidates solely based on instantaneous reward feedback from the current step. However, the information from previously sampled candidates is largely ignored, when the sampling history contain essential signals about how the model distinguishes high-quality proteins from low-quality ones, leading to repetitive design errors and inefficient exploration of the design space. (C3) Over-reliance on high quality design signals: Most existing optimization paradigms focus exclusively on high-reward samples indicating high-quality protein design, but ignore the signals from the low-reward and failed samples, which provide critical decision boundaries in the sparse design landscape of functional proteins.

Driven by the above motivation, we propose a novel framework, **MOMST**, for **M**ulti-**o**bjective protein design via **M**emory-aware **S**elf-contrastive Learning with **T**est-time scaling in diffusion models, for the effective generation of multi-functional protein sequences with balanced performance on conflicting design objectives. MOMST advances a paradigm of **memory-aware test-time scaling**, transforming the generative process into an insight-driven evolutionary process. Specifically, we implement an inference-time alignment strategy that employs weighted summation for local sampling to preserve structural diversity, Pareto Frontier to eliminate dominated solutions and maintain optimal trade-offs, and Pareto sampling for global resampling to maximize the lower bound of all objectives, ensuring a balanced realization of all functional objectives (Addressing C1). We introduce a memory bank to extract generalizable design insights from interaction history. It maintains two components: an elite buffer for historical high-reward samples and a sliding-window exploration buffer for latest exploration. This dual-memory design aligns guidance with the model's progress, effectively preventing repetitive errors (Addressing C2). Building on this powerful experience learner, we utilize Self-contrastive Learning to derive residue-level preference signals by contrasting high-reward sequences against failed attempts. This delineates functional boundaries by suppressing residues linked to instability or functional loss (Addressing C3).

We summarize the contributions of MOMST as follows:

- We propose MOMST, a novel framework that implements Memory-aware Test-time Scaling, recasting the generation process as an insight-driven evolutionary process for multi-objective protein design.

- We develop a closed-loop paradigm combining a Memory Bank with Self-contrastive Learning to distill historical experiences into residue-level preference signals, guiding generation toward high rewards while ensuring structural and biophysical integrity.

- We present an inference-time Pareto alignment strategy that regulates trade-offs between competing multi-objective functions, effectively ensuring the biological viability of designed proteins.

- We evaluate MOMST on both single-objective and challenging multi-objective protein design tasks, and experimental results show that MOMST delivers outstanding performance.

## 2. Related Work

### 2.1. Diffusion Models for Protein Design

Diffusion models have emerged as a powerful paradigm for protein sequence design (Winnifrith et al., 2024; Koh et al., 2025), utilizing either discrete-state processes over amino acid space (Alamdari et al., 2023; Thoutam, 2024; Gruver et al., 2023) or continuous embeddings (Lisanza et al., 2025; Liu et al., 2025; Ingraham et al., 2023). While foundational models like EvoDiff (Alamdari et al., 2023) capture evolutionary distributions, they lack explicit functional steering. ProteinDT (Liu et al., 2025) achieves text-conditioned generation but relies on paired data and struggles with fine-grained biophysical constraints. Similarly, structure-centric models like Chroma (Ingraham et al., 2023) are limited by the computational overhead of 3D coordinate optimization. While (Gruver et al., 2023) introduces inference-time guidance, it fails to navigate complex multi-objective conflicts. Crucially, these approaches often require costly retraining for new target functions and lack the flexibility to integrate multiple constraints (Winnifrith et al., 2024). This gap motivates our use of test-time reward optimization and Pareto alignment to steer pre-trained diffusion models toward diverse multi-objective properties.

### 2.2. Reward-guided Test-time Scaling

Test-time reward optimization has demonstrated strong effectiveness. Most strategies leverage classifier guidance (Dhariwal & Nichol, 2021; Song et al., 2021), incorporating gradients from reward models or classifiers during denoising to steer generation. As reviewed in (Uehara et al., 2025b) have introduced derivative-free methods such as SMC-based guidance (Wu et al., 2023; Dou & Song, 2024; Phillips et al., 2024; Cardoso et al., 2024) or value-based sampling (Li et al., 2024). Derivative-free methods are particularly suited for black-box biophysical constraints common in protein design (e.g., structural metrics like pLDDT or hydrophobicity). Yet, these single-shot generation methods suffer

from irreversible error accumulation, causing sequences to drift from the viable protein manifold due to the lack of an optimization error correction mechanism in the singular denoising trajectory. To mitigate this, Uehara et al. (2025a) propose an iterative refinement approach to optimize complex reward functions. Additionally, evolutionary and genetic algorithms (GAs) (Hie et al., 2022) are widely used to explore the reward landscape via discrete mutations. However, existing refinement frameworks generally operate in a memoryless fashion, failing to extract reasoning insights from the interaction history.

Agents can utilize external memory architectures and self-reflection mechanisms to mitigate repetitive failures (Hu et al., 2025). Ouyang et al. (2025) introduces memory-aware test-time scaling that incorporates agent memory, enabling the model to learn from past experiences and guide future decisions. Drawing inspiration from these works, we transform the stochastic refinement process into a memory-aware optimization paradigm, effectively distilling cumulative design trajectories into residue-level preference signals.

## 3. Methods

### 3.1. Problem Definition

**Notations.** Let $\mathcal{X} = \mathcal{V}^L$ be the space of all possible protein sequences of length $L$, where $\mathcal{V} = \{A_1, \ldots, A_{20}, [\text{M}]\}$ is the vocabulary consisting of the 20 standard amino acids and the mask token $[\text{M}]$. A sequence $x \in \mathcal{X}$ is represented as a vector of residues $(a_1, a_2, \ldots, a_L)$, where each $a_i \in \mathcal{V}$ refers to the amino acid identity at position $i$. We denote $p^{\text{pre}}$ as a pre-trained generative distribution over $\mathcal{X}$. For multi-objective design, we consider $D$ reward functions $\{r_1(x), \ldots, r_D(x)\}$.

**Preliminary.** We formulate multi-objective protein design as a test-time reward optimization problem within a diffusion-based state space $\mathcal{X}$.

■ **Masked Diffusion Models.** The forward process $q(x_t|x_0)$ at step $t$ defines a transition from $x_0$ to $x_t$, where each residue has a probability $1 - \gamma_t$ of being replaced by a mask token $[\text{M}]$ at each step. Formally, the transition probability for a sequence $x_t$ given $x_0$ is defined as: $q_t(x_t|x_0) = \bar{\gamma}_t \delta_{x_t, x_0} + (1 - \bar{\gamma}_t)\delta_{x_t,[\text{M}]}$, where $\delta_{i,j}$ is the Kronecker delta function, $\bar{\gamma}_t = \prod_{i=1}^{t} \gamma_i$ is the cumulative survival probability of a residue remaining unmasked.

The backward process $p_\theta(x_{t-1}|x_t)$, which aims to recover the identities of these masked tokens, is parameterized as:
$$p_\theta(x_{t-1} \mid x_t) =$$
$$\begin{cases} \delta(\cdot = x_t) & \text{if } x_t \neq [\text{M}] \\ \text{Cat}\left(\dfrac{(1 - \bar{\gamma}_{t-1})\delta_{x_{t-1},[\text{M}]} + (\bar{\gamma}_{t-1} - \bar{\gamma}_t)\hat{p}_\theta(x_0 \mid x_t)}{1 - \bar{\gamma}_t}\right) & \text{if } x_t = [\text{M}] \end{cases}$$

where $\hat{p}_\theta(x_0 \mid x_t)$ is a predictor from $x_t$ to $x_0$.

■ **Test-time Reward Optimization.** For reward-guided test-time optimization, which **aims to generate multiple designs that are both natural-like and yield high reward returns**, we sample from the following distribution:

$$p^{(\alpha)} = \operatorname*{argmax}_{p \in \Delta(\mathcal{X})} \mathbb{E}_{x \sim p}[r(x)] - \alpha \text{KL}(p \| p^{\text{pre}})$$
$$= \exp(r(\cdot)/\alpha)p^{\text{pre}}(\cdot)/C \tag{1}$$

where $r(\cdot)$ denotes the reward function and $\alpha$ is a temperature parameter controlling the trade-off between reward maximization and sample native diversity (KL divergence constraint).

In diffusion models, sampling from Eq. 1 is intractable directly. Instead, it is achieved by sequentially sampling from *the soft optimal policy* $\{p_t^\star\}_t$ from $t = T+1$ to $t = 1$, which is defined by

$$p_t^\star(\cdot \mid x_t) \propto \exp\big(v_{t-1}(\cdot)/\alpha\big)p_t^{\text{pre}}(\cdot \mid x_t) \tag{2}$$

Here, $v_t(x_t)$ is the *soft value function*, which acts as a look-ahead function predicting the expected terminal reward at $x_0$ from the current noisy state $x_t$:

$$v_t(x_t) = \alpha \log \mathbb{E}_{x_0 \sim \underbrace{p^{\text{pre}}(x_0 \mid x_t)}_{\text{memoryless}}}\big[\exp\big(r(x_0)/\alpha\big) \mid x_t\big] \tag{3}$$

Crucially, evaluating $v_t$ requires estimating the clean sequence $x_0$ from $x_t$. Standard training-free methods approximate this by relying on the pre-trained model's single-step reconstruction: $v_t(x_t) \approx r(\hat{x}_0(x_t))$, where $\hat{x}_0 \sim p^{\text{pre}}(x_0|x_t)$.

■ **Motivation.** While effective for general generation, this standard approximation suffers from critical limitations for iterative protein design: i) *Lacking memory mechanism:* The fixed pre-trained predictor $p^{\text{pre}}(x_0|x_t)$ lacks a memory mechanism, leading to redundant sampling of proteins with repetitive structural flaws. ii) *Ignoring negative constraints:* Traditional methods lack an explicit penalty for failed regions identified in historical trajectories, which is vital in the sparse functional landscape of proteins. iii) *The difficulty in flexible combination and balance of multiple complex objectives during inference.* In this paper, we propose MOMST to address the above issues.

**Definition 3.1** (**Memory-aware Test-time Scaling for Multi-objective Protein Design**). Given a pre-trained diffusion prior $p^{\text{pre}}(x_0|x_t)$ and an evolving memory bank $\mathcal{M} = \{(x^{(j)}, \mathbf{r}^{(j)})\}$, the objective is to construct a memory-aware policy $p^{\mathcal{M}}$ that solves the following entropy-regularized maximization problem:

$$p^* = \operatorname*{argmax}_{p \in \Delta(\mathcal{X})} \big(\mathbb{E}_{x \sim p^{\mathcal{M}}}[\mathbf{r}(x)] - \alpha \text{KL}(p\|p^{\text{pre}})\big),$$
$$\text{where } \underbrace{p^{\mathcal{M}} = \mathcal{F}(p^{\text{pre}}, \mathcal{M})}_{\text{memory-aware}} \tag{4}$$

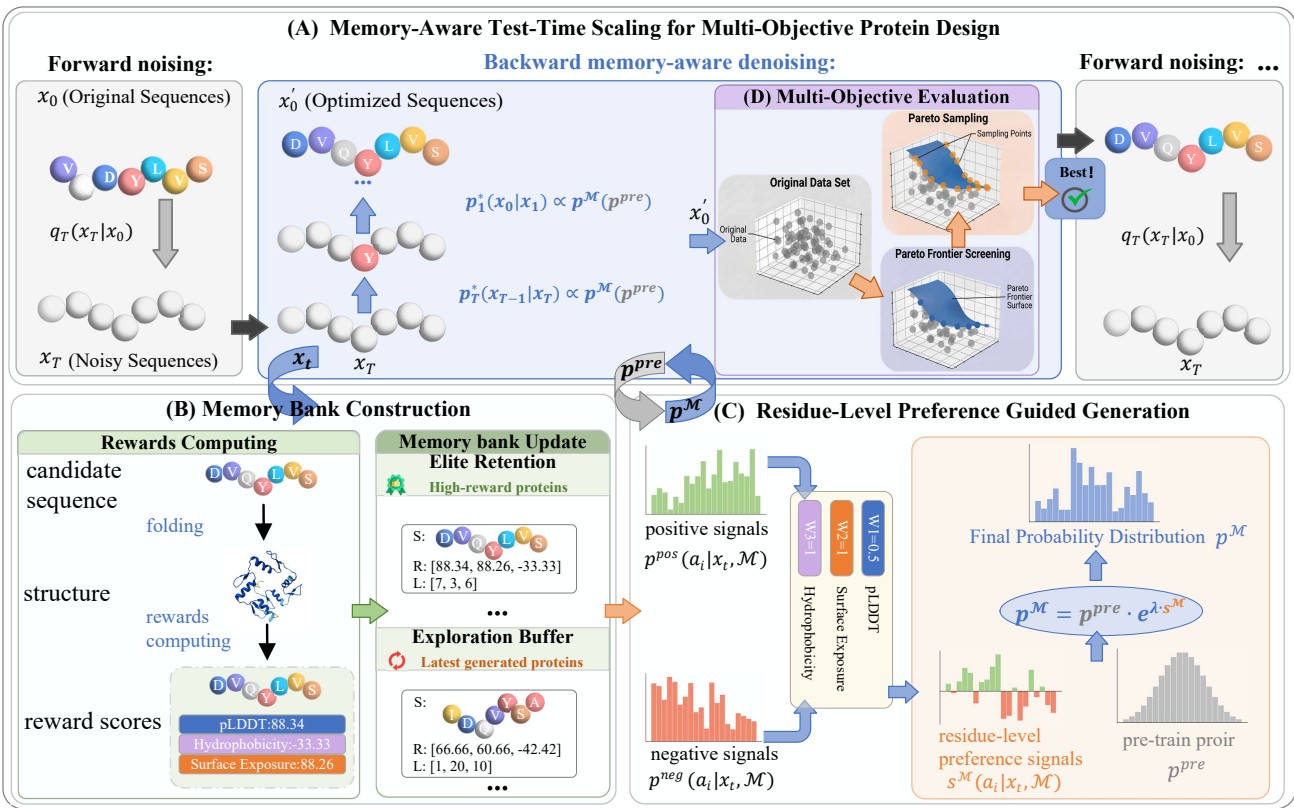

*Figure 1.* The overall process of MOMST is a closed-loop cycle: **(A) Memory-aware Test-time Scaling for Protein Design:** Generating diverse protein candidates as evolutionary foundations while achieving memory-aware iterative optimization of sequences. **(B) Memory Bank Construction:** Aiming to extract generalizable design insights from interaction history. **(C) Residue-level Preference Guided Generation:** The amino acid preference signal is calculated to guide denoising toward Pareto-optimal regions by refining the probability distribution $p^{\text{pre}}$. **(D) Inference-time Pareto Alignment:** Aiming to output multi-objective balanced proteins and re-noised them as starting points for the next evolutionary cycle.

$\mathcal{F}(\cdot)$ is a parameterized modulation function for reconstruction distribution $p^{\text{pre}}(x_0|x_t)$ using guidance from a memory bank.

## 3.2. Overview of the Proposed MOMST

The overall framework of our proposed MOMST for multi-objective protein design is presented in Figure 1. MOMST achieves memory-aware sequence optimization at test-time via an iterative noising and denoising process guided by residue-level preference signals (Figure 1.A). Critically, at each denoising step $t$, we apply a memory-aware soft-optimal strategy to generate protein candidate variants $x_t$ (where seeds are sampled from (Uehara et al., 2025a) in the first iteration). These candidates undergo Multi-Objective Evaluation, where high-performing and diverse samples are then archived in a memory bank $\mathcal{M}$ (Figure 1.B). We derive residue-level preference signals from $\mathcal{M}$, which refine the pre-trained distribution $p^{\text{pre}}$ into a memory-aware target distribution $p^{\mathcal{M}}$ and steering the search toward Pareto-optimal regions while preserving natural-like fidelity (Figure 1.C). Upon completing a denoising trajectory, inference-

time Pareto alignment selects balanced proteins as final outputs or re-noises them to initiate the next evolutionary cycle (Figure 1.D). Detailed algorithms are in Appendix A.1.

## 3.3. Memory Bank Construction

To enable informed decision-making during the iterative refinement process, we implement a Memory Bank ($\mathcal{M}$) that serves as an external, evolving knowledge base. Instead of treating each denoising step as an isolated stochastic event, $\mathcal{M}$ extracts generalizable strategies from interaction history and distills them into structured units to guide generation process of the model at test-time. Each entry is a tuple $E = (S, R, L)$, where $S$ denotes the raw amino acid sequence, $R = \{r_1, r_2, \ldots, r_D\}$ is the multi-objective reward vector, and $L$ represents contextual metadata (iteration, edit length, candidate index).

To maintain a high-quality yet diverse guidance signal, we propose a **hybrid experience consolidation strategy** to update the $\mathcal{M}$. The memory bank is partitioned into two

tiers to balance exploitation and exploration of reasoning-aware memory bank:

■ **Elite Retention (Static Tier):** We reserve 50% of the total memory capacity for storing samples that have performed optimally since the start of the experiment. These "elite" samples provide long-term memory of high-performance patterns, ensuring that the model's forward inference is always based on verified high-reward regions, thereby stabilizing the direction of the search.

■ **Exploration Buffer (Recency Tier):** The remaining 50% of the capacity follows a sliding-window policy, retaining only the most recent interaction history. This enables the framework to sensitively perceive the latest local changes in the search space, prevents the guiding signals from falling into early local optima, and allows the model to learn in real time from recent failures or improvements.

This dual-tiered consolidation ensures that $\mathcal{M}$ maintains a balanced representation of both global optima and local search dynamics. By decoupling structured experience from the underlying generative backbone, this schema transforms non-differentiable biological rewards into a persistent knowledge base.

### 3.4. Residue-level Preference Guided Generation

Protein design in high-dimensional sequence spaces requires navigating a complex landscape where functional motifs are sparse. While traditional reward-guided models focus on learning from successful cases (high rewards), we argue that identifying "what to avoid" is equally critical for efficient convergence. Building on the structured knowledge archived in $\mathcal{M}$, we employ self-contrastive learning on its successful/failed trajectories to extract function-specific, residue-level preference signals, steering generation toward high-performance regions.

For a specific objective $d \in \{1, \ldots, D\}$ and an amino acid position $i$, we define the *Positive Signals* $p^{\text{pos}}(a_i|x_t, \mathcal{M})$ and *Negative Signals* $p^{\text{neg}}(a_i|x_t, \mathcal{M})$. These denote the empirical probability distributions across 20 amino acids, reflecting the residue preferences of high-reward and low-reward sequences.

$$p_d^{\text{pos}}(a_i|x_t, \mathcal{M}) = \frac{\sum_{s \in \mathcal{M}_{high}} \tilde{r}_d(s) \cdot a_{i,s}}{\sum \tilde{r}_d(s) + \epsilon} \quad (5)$$

where $\tilde{r}_d(s)$ denotes the re-scaled reward score of objective $d$, $\epsilon$ is a small constant for numerical stability, $\mathcal{M}_{high}$ denotes sequences above the $k$-th percentile for objective $d$, $a_{i,s}$ denotes the one-hot vector of the 20 standard amino acids at position $i$ of protein sequence $s$. $p^{\text{neg}}(a_i|x_t, \mathcal{M})$ is calculated similarly using $\mathcal{M}_{\text{low}}$.

To handle the complex trade-offs inherent in multi-objective design, we aggregate the contrastive signals across all ob-jectives. We define the total residue-level preference signals $s^{\mathcal{M}}$ as:

$$s^{\mathcal{M}}(a_i|x_t, \mathcal{M}) =$$
$$\sum_{d=1}^{D} w_d \cdot \left(\log p_d^{\text{pos}}(a_i \mid x_t, \mathcal{M}) - \log p_d^{\text{neg}}(a_i \mid x_t, \mathcal{M})\right)$$
$$(6)$$

where $w_d$ represents the relative importance of objective $d$.

Finally, we integrate this preference signal into the reverse diffusion process. The final sampling distribution $p^{\mathcal{M}}$ is modulated by the prior distribution $p^{\text{pre}}$ and the residue-level preference signals:

$$p^{\mathcal{M}} = p^{\text{pre}} \cdot e^{\lambda \cdot s^{\mathcal{M}}} \quad (7)$$

where $\lambda$ is the contrastive scale, employed to modulate the guidance of residue-level preference signals on the pre-trained base distribution during the denoising process. This modulation pulls the model toward residues that historically yielded multi-objective success and pushes it away from those that led to failure.

As iterations proceed, the growing trajectory density within $\mathcal{M}$ provides higher-fidelity empirical distributions for $p^{\text{pos}}$ and $p^{\text{neg}}$. This process allows the model to dynamically reshape the sampling manifold and more efficiently explore the multi-objective landscape than memoryless generation.

### 3.5. Multi-objective Evaluation and Selection

To reconcile the conflicting nature of protein design objectives, we identify the Pareto Front to isolate a set of non-dominated solutions. A candidate is retained if no other solution outperforms it across all metrics simultaneously.

To prevent quality regression, we implement an Elite Preservation strategy. In each iteration, the current non-dominated set is merged with historical elite sequences. Then, we perform global Pareto sampling on this combined pool to select the seeds, which are subsequently re-noised to initiate the next round of refinement until the optimal protein sequence is generated.

## 4. Theoretical Analysis

We present the theoretical analysis of **sampling optimality of memory-aware test-time scaling**.

**Theorem 4.1** Let $p^{(\gamma)}(a_i \mid x_t)$ denote the theoretical optimal conditional distribution at diffusion step $t$ and position $i$, defined as the solution to:

$$p^{(\gamma)} = \arg \max_{p \in \Delta^{19}} \mathbb{E}_{a \sim p}[v(a_i|x_t)] - \gamma \cdot \text{KL}(p \parallel p_{\text{pre}}(\cdot \mid x_t)) \quad (8)$$

*Table 1.* Single-objective protein design results. P50 and P95 denote the median and 95% quantile of the rewards for generated designs. LL denotes the (estimated) per-residue log-likelihood. For both ss-match and cRMSD, we report the mean score of 10 reference proteins randomly chosen (For details on the selection of reference proteins, please refer to Appendix C.2). Best results are in bold.

| MODELS | HYDROPHOBICITY | | | GLOBULARITY | | | SS-MATCH | | | cRMSD | | |
|---|---|---|---|---|---|---|---|---|---|---|---|---|
| | P50↑ | P95↑ | LL↑ | P50↑ | P95↑ | LL↑ | P50↑ | P95↑ | LL↑ | P50↓ | P95↓ | LL↑ |
| SMC | -0.6246 | -0.5274 | -4.9001 | -3.0794 | -2.5398 | -4.8385 | 0.5814 | 0.6894 | -3.9381 | 7.8638 | 5.3611 | -3.6986 |
| SVDD | -0.5386 | -0.3874 | -4.9685 | -3.6063 | -2.5528 | **-4.8157** | 0.7410 | 0.8580 | -3.7843 | 6.8473 | 3.8204 | -3.8034 |
| GA | -0.7080 | -0.6190 | -4.9124 | -4.4090 | -4.0940 | -4.9770 | 0.5901 | 0.6698 | -3.8604 | 8.0961 | 6.2431 | -3.8292 |
| RERD | -0.3812 | -0.2790 | **-4.6717** | -2.4811 | -1.9625 | -4.8464 | 0.9162 | 0.9539 | **-3.6919** | 0.9577 | 0.8496 | -3.6815 |
| **MOMST** | **-0.2242** | **-0.1669** | -4.7504 | **-1.8843** | **-1.6924** | -4.8373 | **0.9542** | **0.9848** | -3.7570 | **0.9109** | **0.7271** | **-3.6775** |

where $v(a_i|x_t)$ is the expected future reward for selecting amino acid $a$ at position $i$ given current state $x_t$.

When the memory bank $\mathcal{M}$ satisfies: i) sufficient size: $|\mathcal{M}| \geq M_{\min}$ where $M_{\min} \to \infty$, ii) high quality: $\rho = |\mathcal{M}^+|/|\mathcal{M}| \geq \rho_{\min} > 0$, and iii) adequate coverage: $\mathcal{M}$ covers the support of the target distribution.

The guided sampling distribution converges to this optimal conditional distribution:

$$\lim_{\substack{|\mathcal{M}|\to\infty \\ \rho\to 1}} \|p^{\mathcal{M}}(\cdot \mid x_t) - p^{(\gamma)}(\cdot \mid x_t)\|_{\text{TV}} = 0 \qquad (9)$$

where $\rho \to 1$ indicates the natural optimization trend of the algorithm (the proportion of high-reward sequences gradually increases) rather than over-reliance on pre-collected successful cases. Consequently, when $p^{\mathcal{M}}$ is sequentially applied across all diffusion steps to modify $p^{\text{pre}}$, the marginal distribution of the generated sequence $p^{\text{gen}}(x_0)$ converges to the global optimal solution of the test-time reward optimization problem:

$$p^* = \arg\max_p \mathbb{E}_{x\sim p}[r(x)] - \gamma\,\text{KL}(p \parallel p^{\text{pre}}) \qquad (10)$$

under standard stability assumptions of the diffusion process. (See Appendix B.1 for full proof, empirical evidence in Appendix C.3.3 also verifies this issue to some extent).

# 5. Experiment

In this section, we comprehensively evaluate the performance of MOMST through multiple single-objective and multi-objective protein design tasks.

**Baselines.** We compare MOMST to single-shot guidance (**SMC** (Wu et al., 2023), **SVDD** (Li et al., 2024)) and memoryless iterative refinement (**GA** (Hie et al., 2022), **RERD** (Uehara et al., 2025a)) baselines, with EvoDiff (Alamdari et al., 2023) as the unconditional base diffusion model (more details in Appendix C.1).

**Reward Function Metrics.** We focus on black-box reward feedback settings, and following existing representative works in protein design (Hie et al., 2022; Watson

et al., 2023; Ingraham et al., 2023), we consider structural-property reward functions that take generated sequences as input: **hydrophobicity** measures the water-repellent property of a molecule, **globularity** reflects how closely the structure resembles a globular shape, **ss-match** is the mean matching probability across all residues between the predicted and reference secondary structures, **cRMSD** is the constrained root mean square deviation against the reference backbone structure after structural alignment, **surface exposure** quantifies the structural exposure of protein fragments, enabling to assess potential biological functional activity, **pLDDT** predicts the local accuracy of a protein structure. **Notably, to facilitate algorithm optimization and comparison, most reward metrics follow a higher-is-better strategy (e.g., hydrophobicity is negated from its original value).** For more details, refer to Appendix C.2.

Beyond optimization reward objectives, we evaluate design reliability using two auxiliary metrics. **LL** (Log-Likelihood) measures sequence naturalness to verify that the generated protein does not drift from the pre-trained prior, while **pTM** assesses global structural reliability and overall foldability.

## 5.1. MOMST for Single-objective Protein Design

We first evaluate MOMST on single-objective design tasks, focusing on the efficacy of the memory-aware learning mechanism. As shown in Table 1, **MOMST consistently outperforms all baseline models across all reward metrics.** Single-shot methods such as SMC and SVDD attempt to bias the generation toward high-reward regions in a derivative-free manner, they are inherently limited by their inability to perform sequential error correction. Although iterative strategies like RERD utilize alternating noising and reward-guided denoising steps to progressively correct errors, they operate in a memoryless fashion, often encountering repetitive optimization traps. Results demonstrate that MOMST distills cumulative design trajectories into residue-level preference signals to maximize functional rewards while maintaining sequence naturalness (reflected by the LL scores), effectively preventing repetitive errors and drift from the natural biophysical manifold.

*Table 2.* Dual-objective protein design results. The hydrophobicity-pLDDT combination ensures core-driven folding stability, while the globularity-pLDDT combination provides structural confidence in a compact sphere for stable scaffold design. Best results are in bold.

| MODELS | HYDROPHOBICITY & PLDDT | | | | | GLOBULARITY & PLDDT | | | | |
|---|---|---|---|---|---|---|---|---|---|---|
| | P50↑ | P95↑ | P50↑ | P95↑ | LL↑ | P50↑ | P95↑ | P50↑ | P95↑ | LL↑ |
| SMC | -0.6871 | -0.6671 | 0.4287 | 0.6271 | -5.2382 | -3.5201 | -2.9851 | 0.3606 | 0.6216 | -4.8188 |
| SVDD | -0.6347 | -0.5608 | 0.9152 | 0.9510 | -5.2675 | -3.6062 | -2.5303 | 0.4022 | 0.6487 | -4.8305 |
| GA | -0.6809 | -0.5699 | 0.4058 | 0.4514 | **-4.8556** | -4.1058 | -3.4443 | 0.3087 | 0.3865 | -4.9417 |
| RERD | -0.5772 | -0.5734 | 0.9563 | 0.9631 | -4.9577 | -2.4370 | -2.1871 | 0.5084 | 0.5365 | -4.9101 |
| **MoMST** | **-0.5556** | **-0.5406** | **0.9657** | **0.9704** | -5.1972 | **-1.7356** | **-1.6917** | **0.7176** | **0.7498** | **-4.7392** |

*Table 3.* Triple-objective protein design results. The hydrophobicity-surface exposure-pLDDT combination suits therapeutic protein design, ensuring high structural stability, solubility, and reduced aggregation-mediated immunogenic risks. The hydrophobicity-globularity-pLDDT combination enables *de novo* design of compact protein scaffolds and nanocages by enforcing a robust hydrophobic core in a strictly defined, high-confidence spherical geometry. Triple-objective tasks also maintain reasonably high LL score (refer to Table 4). Best results are in bold.

| MODELS | HYDROPHOBICITY & SURFACE EXPOSURE & PLDDT | | | | | | HYDROPHOBICITY & GLOBULARITY & PLDDT | | | | | |
|---|---|---|---|---|---|---|---|---|---|---|---|---|
| | P50↑ | P95↑ | P50↑ | P95↑ | P50↑ | P95↑ | P50↑ | P95↑ | P50↑ | P95↑ | P50↑ | P95↑ |
| SMC | -0.7066 | -0.6014 | 0.7797 | 0.8338 | 0.4853 | 0.6031 | -0.6924 | -0.6095 | -3.6002 | -2.5165 | 0.4274 | 0.5391 |
| SVDD | -0.6673 | -0.6191 | 0.7164 | 0.7499 | 0.8826 | **0.9358** | -0.7152 | -0.3758 | -3.5680 | -2.6504 | 0.4351 | 0.6448 |
| GA | -0.6344 | -0.5668 | 0.8220 | 0.8437 | 0.6296 | 0.6446 | -0.6867 | -0.6054 | -4.5079 | -3.8141 | 0.3687 | 0.4150 |
| RERD | -0.5924 | -0.5815 | 0.7521 | 0.7634 | 0.8431 | 0.8534 | -0.2857 | -0.2857 | **-1.9976** | **-1.9976** | 0.4187 | 0.4716 |
| **MoMST** | **-0.3808** | **-0.3399** | **0.8462** | **0.8764** | **0.8872** | 0.8965 | **-0.2466** | **-0.2270** | -3.2023 | -3.0847 | **0.7657** | **0.7969** |

## 5.2. MoMST for Multi-objective Protein Design

Multi-objective protein design challenges require navigating high-dimensional trade-offs between conflicting biophysical constraints. Tables 2 and 3 illustrate the true strength of MoMST in achieving superior balance within these complex landscapes. Single-shot methods, e.g., SMC and SVDD, often fail to accurately approximate optimal guidance directions under multiple objective constraints, as limited sampling makes it difficult to locate the narrow intersection of multiple high-reward regions. Iterative strategies, such as GA and RERD, which often rely on naive scalarization (e.g., weighted summation of rewards), frequently experience objective collapse, where optimizing one property leads to a sharp decline in another. This is most evident in the triple-objective task (hydrophobicity, globularity, and pLDDT), where RERD achieves high globularity scores but suffers from a critical deficiency in pLDDT ($< 0.7$), rendering the generated proteins highly unreliable. **In contrast, MoMST provides the only model that reliably generates proteins whose predicted structures are well-folded and structurally plausible (pLDDT $> 0.7$) across all multi-objective scenarios.** In the triple-objective task involving hydrophobicity, surface exposure, and pLDDT, MoMST outperforms all baselines. These results validate the effectiveness of MoMST, which maximizes functional rewards via memory-aware test-time scaling and resolves conflicting multi-functional biological requirements through inference-time Pareto alignment. Meanwhile, the high LL scores in Table 4 further demonstrate that our proposed MoMST can lead to superior reward optimization while maintain-

*Table 4.* LL results for Triple-objective protein design, where hydro., glob. and surfexp. denote hydrophobicity, globularity and surface exposure, respectively. Best results are in bold.

| MODELS | HYDRO. & SURFEXP. & PLDDT | HYDRO. & GLOB. & PLDDT |
|---|---|---|
| | LL↑ | LL↑ |
| SMC | -4.8612 | -4.8853 |
| SVDD | -4.8789 | -5.0947 |
| GA | -4.9072 | -4.5962 |
| RERD | -4.9872 | **-4.3920** |
| **MoMST** | **-4.6933** | -4.9344 |

ing biologically plausible sequences under multi-objective tasks. For a comprehensive evaluation of MoMST's multi-objective performance, we report statistical means and standard deviations (Figure 2), which confirm its near-universal superiority and superior Pareto alignment.

Furthermore, to assess global structural reliability, we provide additional results for pTM (in Table 5). The pTM scores of MoMST consistently outperform all baselines across multi-objective combinations, successfully surpassing the accepted foldability threshold of 0.5. Notably, MoMST achieves a pTM of 0.7702 in the double-objective task (Hydrophobicity and pLDDT), and a pTM of 0.5454 in the triple-objective task (Hydrophobicity, Surface Exposure, and pLDDT), which is nearly four times that of RERD (0.1446). This demonstrates that MoMST effectively prevents reward hacking, ensuring that the pursuit of high functional rewards does not compromise the global foldability and structural credibility of the designed proteins. We also

*Table 5.* pTM results for multi-objective protein design, where hydro., glob. and surfexp. denote hydrophobicity, globularity and surface exposure, respectively. Best results are in bold.

| MODELS | HYDRO. & PLDDT PTM↑ | GLOB. & PLDDT PTM↑ | HYDRO. & SURFEXP. & PLDDT PTM↑ | HYDRO. & GLOB. & PLDDT PTM↑ |
|---|---|---|---|---|
| SMC | 0.2691 | 0.2245 | 0.2311 | 0.3084 |
| SVDD | 0.6672 | 0.1944 | 0.2177 | 0.6274 |
| GA | 0.2228 | 0.1941 | 0.2111 | 0.3670 |
| RERD | 0.6383 | 0.1602 | 0.1446 | 0.5645 |
| **MoMST** | **0.7702** | **0.5257** | **0.5454** | **0.6439** |

*Table 6.* Ablation study of the MoMST model, where hydro., glob. and surfexp. denote hydrophobicity, globularity and surface exposure, respectively. Best results are in bold.

| MODELS | GLOB. & PLDDT | | | | HYDRO. & SURFEXP. & PLDDT | | | | | |
|---|---|---|---|---|---|---|---|---|---|---|
| | P50↑ | P95↑ | P50↑ | P95↑ | P50↑ | P95↑ | P50↑ | P95↑ | P50↑ | P95↑ |
| **MoMST** | **-1.7356** | **-1.6917** | 0.7176 | 0.7498 | -0.3808 | -0.3399 | **0.8462** | **0.8764** | **0.8872** | **0.8965** |
| W/O REASONING-AWARE MEMORY BANK | -3.5537 | -3.2303 | 0.3812 | 0.4764 | **-0.1961** | **-0.1765** | 0.7290 | 0.7377 | 0.8815 | 0.8920 |
| W/O SELF-CONTRASTIVE LEARNING | -3.3601 | -3.3420 | **0.8710** | **0.8846** | -0.5812 | -0.5401 | 0.8441 | 0.8522 | 0.8689 | 0.8739 |
| W/O INFERENCE-TIME PARETO ALIGNMENT | -2.8370 | -2.6022 | 0.4824 | 0.6187 | -0.6931 | -0.6183 | 0.7701 | 0.7780 | 0.8314 | 0.8627 |

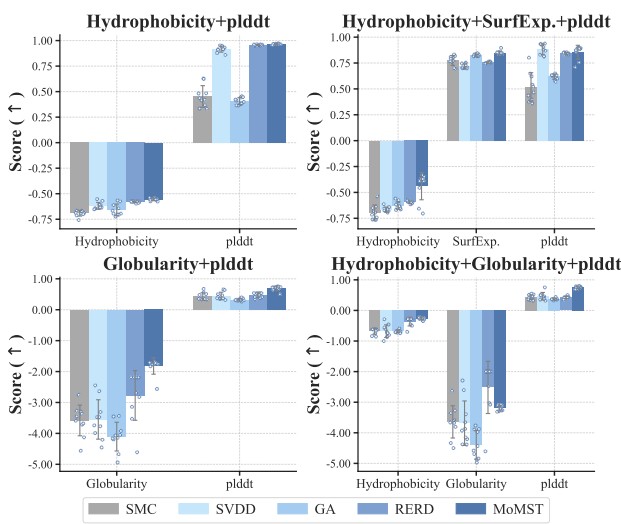

*Figure 2.* Mean scores of each reward in multi-objective design.

assessed the average rank (AvgRank) across all tasks and the diversity performance of MoMST, with MoMST ranking first (see Appendix C.3.1).

### 5.3. Ablation Study

Ablation studies in Table 6 confirm the critical contribution of each MoMST component. Removing *the Reasoning-Aware Memory Bank replaces the Recency Tier with a static reward-based policy*, leading to stale early-iteration negative samples, distorted residue-level preference signals and impaired avoidance of repetitive optimization errors (e.g., lower glob. and pLDDT). Although hydrophobicity marginally rises, this comes at the cost of reduced surfexp. (P95: 0.8764 → 0.7377) and pLDDT. Furthermore, removing *Self-contrastive Learning ignores failure sam-*

*ples*, which results in marked performance declines. This validates our core motivation that failure cases provide indispensable guidance for complex biophysical manifolds, whereas success-only signals are inadequate. Finally, *removing the Inference-Time Pareto Alignment* demonstrates its key role in balancing the functions in multi-objective protein design. This removal severely degrades multi-objective balance, evidenced by a sharp drop in structural reliability (with the pLDDT at P50 from 0.7176 to 0.4824 in the glob. & pLDDT). See Appendix C.3.2 and C.3.3 for additional parameter analyses.

### 5.4. Case Study

To demonstrate the precision of MoMST, Figure 3 showcases its performance across diverse design scenarios. Specifically, (A-C) present generated protein structures from single-objective and multi-objective tasks, illustrating the broad applicability of MoMST. This is followed by (D-E), a detailed visualization of optimized surface hydrophobicity focusing on multi-objective design. As illustrated, MoMST effectively regulates surface hydrophobicity compared to the RERD. While generated proteins of RERD exhibit dense hydrophobic patches (yellow regions), MoMST produces structures with significantly reduced hydrophobic exposure (the cyan regions are expanding), thereby mitigating aggregation risks while maintaining structural stability.

Figure 4 visualizes the optimization trajectories across iterations. MoMST achieves a steady and superior increase in cumulative rewards compared to GA and RERD. This trend demonstrates that our memory-aware scaling effectively navigates the complex multi-objective landscape, consistently converging toward high-quality proteins with optimized biological efficacy.

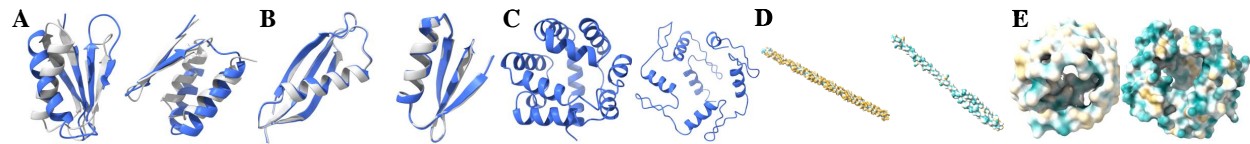

*Figure 3.* Results of MOMST for multi-objective protein design. (**A**) Generated proteins (blue) for optimizing **ss-match** are shown, silver denotes target secondary structures. The left (6NJF) has an ss-match score of 1.0, and the right one (XX_run1_0254_0003) 0.98. (**B**) Generated proteins (blue) for **cRMSD** optimization are shown, silver denotes the target backbone structures. The left (r15_96_TrROS_Hall) has a cRMSD score of 0.36, and the right (EHEE_rd1_0101) 0.35. (**C**) Left: Protein generated with **globularity and pLDDT** as objectives. Right: Proteins generated with **hydrophobicity, globularity and pLDDT** as objectives. (**D**) Surface hydrophobicity mapping (cyan: polar; yellow: hydrophobic) in optimizing **hydrophobicity, surface exposure and pLDDT**: MOMST (right) reduces hydrophobic patch density vs. RERD (left), mitigating aggregation risk while ensuring structural stability. (E) Surface hydrophobicity mapping (cyan: polar; yellow: hydrophobic) in optimizing **hydrophobicity, globularity and pLDDT**: MOMST (right) reduces hydrophobic patch density vs. RERD (left), mitigating aggregation risk while ensuring structural stability.

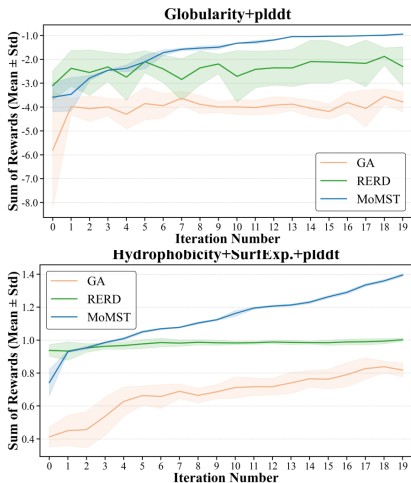

*Figure 4.* Optimization trajectories of rewards in multi-objective protein design (rewards aggregated for comparison).

## 6. Conclusion

In this paper, we propose MOMST, a memory-aware test-time scaling framework that enables flexible combination of objectives for multi-objective protein design without retraining. By introducing a memory bank, we overcome the repetitive error accumulation inherent in memoryless reward-guided diffusion. Crucially, MOMST leverages self-contrastive learning to distill success and failure trajectories into residue-level signals, steering generation toward high rewards while strictly maintaining the natural biophysical manifold. Integrating inference-time Pareto alignment, MOMST ensures a robust balance across conflicting objectives. Evaluations on single and multi-objective tasks demonstrate that MOMST provides a reliable and flexible paradigm for multi-objective protein design. Future work will investigate its scalability across diverse model architectures and more complex biological molecular design landscapes.

## Acknowledgements

The authors thank the anonymous reviewers for their valuable suggestions. Yi Li was supported by the Fundamental Research Funds for the Central Universities (No. DUT25YG238). Xin Zheng acknowledged support from the 2025 NVIDIA Academic Grants Program in the field of protein design.

## Impact Statement

This study proposes a novel memory-aware diffusion-based test-time reward optimization method for multi-objective protein design. This method leverages a reasoning-aware memory bank and a self-contrastive learning strategy to guide the generation. In addition, a Pareto alignment mechanism is introduced to ensure a balance among multiple objective functions. Our work aims to enhance the model's capability to optimize target rewards during test-time by learning effective reward signals from iterative history, enabling adaptation to evolving functional objectives without incurring the high cost of retraining, and this holds potential significance for fields such as drug discovery and biomolecular engineering. When used responsibly, such models can accelerate the advancement of protein engineering by optimizing the generation quality of multi-objective proteins.

At the same time, we recognize that if misapplied, protein generation models, particularly those optimized for specific reward functions, may be subject to abuse, such as the design of harmful biological agents. However, our work is computational in nature, intended to serve scientific research, and does not constitute evidence of real-world functionality or safety; furthermore, we emphasize the importance of the responsible deployment of biotechnologies and alignment with ethical guidelines. Overall, our contributions align with the broader objectives of machine learning methodologies, and we foresee no direct concerns beyond the universal ethical issues associated with generation models.

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

# A. Model Details

## A.1. Pseudo Code

Algorithm 1 provides the pseudo code for this paper.

---

**Algorithm 1** MOMST: Multi-objective Protein Design via Memory-aware Test-time Scaling

---

**Require:** Pre-trained diffusion model $p_\theta$, reward model $\mathcal{R}$, batch size $B$, target protein length $L$, iteration count $N$, noise level $T$ (editable sequence length), candidate number $C$, memory size $M_{\text{size}}$, rewards number $D$, rewards weights w
**Ensure:** Optimized protein sequences $\{\mathbf{s}^*\}$
 1: Initialize memory bank $\mathcal{M} \leftarrow \emptyset$
 2: Initialize masked sequences $\mathbf{s}^{(0)} \in \mathbb{Z}^{B \times L} \leftarrow [\text{[MASK]}]^{B \times L}$
 3: Define temperature schedule: $\tau(n) = 10.0 \cdot (1 - n/N) + 1.0 \cdot (n/N)$
 4: **for** $n = 0$ to $N - 1$ **do**
 5: $\quad$ $\mathbf{s} \leftarrow \mathbf{s}^{(n)}$
 6: $\quad$ Set current temperature $\tau_n \leftarrow \tau(n)$
 7: $\quad$ **if** $n = 0$ **then**
 8: $\quad\quad$ Editable sequence length $T \leftarrow L$
 9: $\quad$ **else**
10: $\quad\quad$ Editable sequence length $T$
11: $\quad$ **end if**
12: $\quad$ **for** $t = 0$ to $T - 1$ **do**
13: $\quad\quad$ Initialize candidate list $\mathcal{C} \leftarrow [\,]$
14: $\quad\quad$ **for** $c \leftarrow 0$ to $C - 1$ **do**
15: $\quad\quad\quad$ For each $b$, pick editable position $i \sim \text{Uniform}(0, L)$
16: $\quad\quad\quad$ Predict logits: $\mathbf{p} \leftarrow p_\theta(\mathbf{s}, t)$
17: $\quad\quad\quad$ **for** $b \leftarrow 0$ to $B - 1$ **do**
18: $\quad\quad\quad\quad$ Get base distribution: $p_{i_b}^{pre} \leftarrow \text{Softmax}(\mathbf{p}_{i_b}[p_{i_b}, : 20]/\tau_n)$
19: $\quad\quad\quad\quad$ $\mathbf{p}_{i_b}^{\mathcal{M}} \leftarrow$ Residue-Level Preference Extraction$(\mathbf{p}_{i_b}^{\text{pre}}, \mathcal{M}, D, \mathbf{w})$ $\quad\triangleright$ Please refer to Algorithm 2.
20: $\quad\quad\quad\quad$ Normalize $p_{i_b}^{\mathcal{M}}$
21: $\quad\quad\quad\quad$ Sample amino acid: $a_i \sim \text{Multinomial}(p_i^{\mathcal{M}})$
22: $\quad\quad\quad\quad$ Update candidate:
23: $\quad\quad\quad\quad$ Set $\mathbf{s}_{b,i_b}^{\text{cand}} \leftarrow \mathbf{a}_b$ for all $b$
24: $\quad\quad\quad$ **end for**
25: $\quad\quad\quad$ $\mathcal{C}.append(\mathbf{s}^{\text{cand}})$
26: $\quad\quad$ **end for**
27: $\quad\quad$ Evaluate all candidates: for each $c$, compute multiple rewards $r_{b,c}^d = \mathcal{R}(\mathbf{s}_c^{\text{cand}})_b \in \mathbb{R}^D$
28: $\quad\quad$ Compute weighted score: $s_{b,c} \leftarrow \sum_{d=1}^D w_d \cdot \text{RankNorm}(\mathbf{r}_{b,c}^d)$
29: $\quad\quad$ Select best index per sample: $c_b^* \leftarrow \arg\max_c r_{b,c}$
30: $\quad\quad$ Update $\mathbf{s} \leftarrow \mathcal{C}[c_b^*] \quad \forall b \in \{0, \dots, B - 1\}$
31: $\quad$ **end for**
32: $\quad$ Update memory bank: UpdateMemoryBank$(\mathcal{M}, n, t, \mathbf{s}, \mathbf{r}, \mathbf{w}, M_{\text{size}})$ $\quad\triangleright$ Please refer to Algorithm 3.
33: $\quad$ $\mathbf{s}^{(n+1)} \leftarrow$ Multi-objective Evaluation and Selection$(s, r_{b,c}^d)$ $\quad\triangleright$ Please refer to Algorithm 4.
34: **end for**
35: Save final memory bank $\mathcal{M}$ and final sequences $\{\mathbf{s}^*\}$

---

---

**Algorithm 2** Residue-level Preference Extraction

---

**Require:** Memory bank $\mathcal{M}$, position $i$, objective dimension $d \in \{1, \ldots, D\}$, positive and negative sample thresholds: $\tau_{\text{high}}, \tau_{\text{low}}$, contrastive scale $\lambda$

**Ensure:** $p^{\mathcal{M}}$

1: **if** $\mathcal{M} = \emptyset$ **then**
2:      Return: uniform distributions
3: **end if**
4: Extract raw rewards for objective $d \in D$: $\mathbf{r} \leftarrow [\text{item.rewards}[d] \mid \text{item} \in \mathcal{M}]$
5: Normalize via rank: $\hat{\mathbf{r}} \leftarrow \text{RankNormalize}(\mathbf{r})$
6: **for** each objective dimension $d$ **do**
7:      **for** each item in $\mathcal{M}$ **do**
8:          $a \leftarrow \text{item.pos\_probs}[i]$ {The one-hot vector of the 20 standard amino acids at position $i$}
9:          **if** $\hat{\mathbf{r}} \geq \tau_{\text{high}}$ **then**
10:              $p^{\text{pos}} \leftarrow p^{\text{pos}} + \hat{\mathbf{r}} \cdot a$
11:          **else if** $\hat{\mathbf{r}} \leq \tau_{\text{low}}$ **then**
12:              $p^{\text{neg}} \leftarrow p^{\text{neg}} + \hat{\mathbf{r}} \cdot a$
13:          **end if**
14:      **end for**
15:      Normalize $p^{\text{pos}}, p^{\text{neg}}$
16:      $s_d \leftarrow \log p^{\text{pos}} - \log p^{\text{neg}}$
17:      $s^{\mathcal{M}} += w_d \cdot s_d$
18: **end for**
19: $p^{\mathcal{M}} = p^{\text{pre}} \cdot e^{\lambda \cdot s^{\mathcal{M}}}$
20: Return $p^{\mathcal{M}}$

---

**Algorithm 3** UpdateMemoryBank: Reasoning-aware Memory Bank Update

---

**Require:** Memory bank $\mathcal{M}$, iteration $n$, edit step $e$, candidate id $c$, sequence $\mathbf{s}$, raw rewards $\mathbf{r} \in \mathbb{R}^D$, rewards weights $\mathbf{w}$, max size $M_{\text{size}}$

**Ensure:** Updated memory bank $\mathcal{M}$

1: **if** $\mathbf{s} \in \{\text{item.sequence} \mid \text{item} \in \mathcal{M}\}$ **then**
2:      Return {skip duplicates}
3: **end if**
4: Encode positional one-hot: $\text{pos\_probs} \in \mathbb{R}^{|\mathbf{s}| \times 20}$
5: Add new entry:

$$\text{entry} \leftarrow \{\text{location\_info} : (n, e, c), \text{ sequence} : \mathbf{s}, \text{ rewards} : \mathbf{r}, \}$$

6: $\mathcal{M}.append(\text{entry})$
7: {Global rank normalization across all entries}
8: Extract all rewards: $\mathbf{r} \leftarrow [\text{item.rewards} \mid \text{item} \in \mathcal{M}] \in \mathbb{R}^{B \times D}$
9: **for** each objective $d = 1$ to $D$ **do**
10:      $\tilde{\mathbf{r}}_{:,d} \leftarrow \text{RankNormalize}(\mathbf{r}_{:,d})$
11: **end for**
12: Compute final scores: $\mathbf{sum\_score} \leftarrow \tilde{\mathbf{r}} \cdot \mathbf{w}$
13: *// Hybrid consolidation: Elite Buffer + Exploration Buffer*
14: $K_{\text{elite}} \leftarrow \lfloor 0.5 \cdot M_{\text{size}} \rfloor$
15: Sort $\mathcal{M}$ by `sum_score` descending
16: $\text{elites} \leftarrow \mathcal{M}[0 : K_{\text{elite}}]$
17: $\text{remaining} \leftarrow \mathcal{M}[K_{\text{elite}} :]$
18: $\text{recent} \leftarrow \text{last } (M_{\text{size}} - K_{\text{elite}})$ of remaining
19: $\mathcal{M} \leftarrow \text{elites} \cup \text{recent}$
20: Re-sort $\mathcal{M}$ by `sum_score` ascending (for future percentile queries)

---

---

**Algorithm 4** Multi-objective Evaluation and Selection: Inference-time Pareto Alignment

---

**Require:** Current samples $\mathbf{s}$, raw rewards $\mathbf{r} \in \mathbb{R}^{B \times D}$, weights $\mathbf{w}$, batch size $B$, memory bank $\mathcal{M}$
**Ensure:** Resampled sequences $\mathbf{s}'$
 1: Compute ranked rewards: $\tilde{\mathbf{r}}_{:,d} \leftarrow \text{RankNormalize}(\mathbf{r}_{:,d})$ for all $d$
 2: Identify Pareto front indices: $\mathcal{I}_{\text{pf}} \leftarrow \text{ParetoFilter}(\tilde{\mathbf{r}})$
 3: $\mathbf{s}_{\text{pf}} \leftarrow \mathbf{s}[\mathcal{I}_{\text{pf}}]$, $\mathbf{r}_{\text{pf}} \leftarrow \mathbf{r}[\mathcal{I}_{\text{pf}}]$
 4: // *Elite preservation*
 5: $\mathcal{M}_{\text{top}} \leftarrow \text{Top-}K$ items from $\mathcal{M}$ by sum_score
 6: $\mathbf{s}_{\text{top}} \leftarrow [\text{item.sequence for item in } \mathcal{M}_{\text{top}}]$
 7: $\tilde{\mathbf{r}}_{\text{top}} \leftarrow [\text{item.rewards for item in } \mathcal{M}_{\text{top}}]$
 8: Combine pools:

$$\mathbf{s}_{\text{pool}} \leftarrow \text{Concat}(\mathbf{s}_{\text{pf}}, \mathbf{s}_{\text{top}})$$

$$\tilde{\mathbf{r}}_{\text{pool}} \leftarrow \text{Concat}(\tilde{\mathbf{r}}_{\text{pf}}, \mathbf{r}_{\text{top}})$$

 9: Compute robust score per sample:

$$r_{pareto} \leftarrow \min_d \left( \text{RankNormalize}(\tilde{\mathbf{r}}_{\text{pool},:,d}) \right)$$

10: Perform resampling: $\forall b \in [B]$, sample $\mathbf{s}' \sim \sum_{b=1}^{B} w_b \delta_{\mathbf{s}',b}$, $w_b = \frac{\exp(r_{pareto}(\mathbf{s}_{\mathbf{pool},\mathbf{b}})/\alpha)}{\sum_B \exp(r_{pareto}(\mathbf{s}_{\mathbf{pool},\mathbf{B}}))/\alpha}$
11: Return $\mathbf{s}'$

---

# B. Proofs

### B.1. Proof of Theorem 4

**Proof.** We prove Theorem 4.1 through five key steps:

*// Step 1: Definition of the Single-Step Optimization Objective*

At diffusion step $t$ and position $i$, we define the single-step optimization objective to maximize the expected future reward (value function) while preserving fidelity to the pre-trained distribution $p^{\mathrm{pre}}$:

$$\max_{p \in \Delta_{19}} \mathbb{E}_{a \sim p} \left[ v(a_i; x_t) \right] - \gamma \cdot \mathrm{KL} \left( p \parallel p^{\mathrm{pre}}(\cdot \mid x_t) \right) \tag{11}$$

where the value function $v(a_i; x_t)$ is defined as:

$$v(a_i; x_t) = \mathbb{E}_{x_0 \sim p^{\mathrm{pre}}(\cdot \mid x_t, a_i = a)} \left[ r(x_0) \right]$$

which denotes the expected final reward of the generated sequence $x_0$ given the current diffusion state $x_t$ and candidate amino acid $a$ at position $i$.

This objective balances two core goals of protein sequence optimization: 1. **Reward Maximization**: The first term encourages the selection of amino acids that lead to higher final functional rewards of the protein sequence. 2. **Pre-trained Fidelity Preservation**: The KL divergence term regularizes the optimized distribution $p$ to avoid excessive deviation from the pre-trained distribution $p^{\mathrm{pre}}$, which encodes the natural patterns of native protein sequences.

*// Step 2: Derivation of the Theoretical Optimal Distribution*

**Proposition**. *The constrained optimization problem in Step 1 has a unique optimal distribution $p^{(\alpha)}$, which can be derived using the method of Lagrange multipliers.*

*Proof.* First, the optimization is subject to the probability simplex constraint $\sum_a p(a) = 1$ (the sum of probabilities over all candidate amino acids equals 1). We introduce a Lagrange multiplier $\mu$ to enforce this constraint, and construct the Lagrangian function as follows:

$$\mathcal{L}(p, \mu) = \sum_a p(a) v(a) - \gamma \sum_a p(a) \log \frac{p(a)}{p^{\mathrm{pre}}(a)} + \mu \left( 1 - \sum_a p(a) \right) \tag{12}$$

Then, to find the extremum of the Lagrangian, we take the partial derivative of $\mathcal{L}$ with respect to $p(a)$ and set the result to zero:

$$\frac{\partial \mathcal{L}}{\partial p(a)} = v(a) - \gamma \left( \log \frac{p(a)}{p^{\mathrm{pre}}(a)} + 1 \right) - \mu = 0 \tag{13}$$

*Derivation Note*: The partial derivative of the KL divergence term $\sum_a p(a) \log \frac{p(a)}{p^{\mathrm{pre}}(a)}$ with respect to $p(a)$ is $\log \frac{p(a)}{p^{\mathrm{pre}}(a)} + 1$, which is obtained by applying the product rule of differentiation for the term $p(a) \log p(a)$ and the constant rule for $p(a) \log p^{\mathrm{pre}}(a)$ (where $p^{\mathrm{pre}}(a)$ is independent of $p(a)$).

Rearrange Equation (13) to isolate the logarithmic term:

$$\log \frac{p(a)}{p^{\mathrm{pre}}(a)} = \frac{v(a) - \gamma - \mu}{\gamma}$$

Exponentiate both sides of the equation to eliminate the logarithm:

$$p(a) = p^{\mathrm{pre}}(a) \cdot \exp \left( \frac{v(a)}{\gamma} - \frac{\gamma + \mu}{\gamma} \right)$$

To simplify the expression, we define two constants: $\beta = \frac{1}{\gamma}$ (temperature coefficient controlling the strength of reward weighting) and $C = \frac{\gamma + \mu}{\gamma}$ (a constant absorbing the Lagrange multiplier and regularization coefficient). Substituting these constants gives:

$$p(a) = p^{\mathrm{pre}}(a) \cdot \exp \left( \beta \cdot v(a) - C \right)$$

Finally, enforce the probability simplex constraint $\sum_a p(a) = 1$ by normalizing the unnormalized distribution. The unique theoretical optimal distribution $p^{(\alpha)}$ is thus:

$$p^{(\alpha)}(a) = \frac{p^{\text{pre}}(a) \cdot \exp\left(\beta \cdot v(a)\right)}{\sum_{a' \in \mathcal{A}} p^{\text{pre}}(a') \cdot \exp\left(\beta \cdot v(a')\right)} \tag{14}$$

where $\mathcal{A}$ denotes the full set of candidate amino acids. Therefore, there exists an optimal distribution $p^{(\alpha)}$ that maximizes the expected reward while remaining close to the pre-trained distribution.

*// Step 3: Proof of Linear Relationship Between $s^{\mathcal{M}}$ and $v(a)$*

**Proposition**. *When the memory bank $\mathcal{M}$ is sufficiently large and high-quality, the total residue-level preference signals $s^{\mathcal{M}}(a)$ is a linear function of the true reward function $v(a)$.*

*Proof.* First, assuming the samples in the memory bank are independent and identically distributed, the empirical distributions of positive and negative samples converge to their respective true conditional distributions as $|\mathcal{M}| \to \infty$ (by the Strong Law of Large Numbers):

$$p^{\text{pos}}(a_i, \mathcal{M}) \to p^{\text{true}}(a \mid v(x) > \tau_{\text{high}}, x_{\backslash i})$$
$$p^{\text{neg}}(a_i, \mathcal{M}) \to p^{\text{true}}(a \mid v(x) \le \tau_{\text{low}}, x_{\backslash i})$$

where $x_{\backslash i}$ denotes the protein sequence excluding the $i$-th position, $\tau_{\text{high}}$ is the high reward threshold for positive samples, and $\tau_{\text{low}}$ is the low reward threshold for negative samples.

In statistical mechanics and machine learning for sequence optimization, it is a standard assumption that reward-conditioned distributions belong to the exponential family. Thus, for the true conditional distributions, we have:

$$p^{\text{true}}(a \mid v(x) > \tau) \propto \exp(k \cdot v(a)) \cdot p^{\text{pre}}(a)$$

$$p^{\text{true}}(a \mid v(x) \le \tau) \propto \exp(-k \cdot v(a)) \cdot p^{\text{pre}}(a)$$

where $k > 0$ is a constant related to the variance of the reward distribution, and $\propto$ denotes proportionality.

Then, we take the ratio of the two conditional distributions to eliminate the pre-trained distribution $p^{\text{pre}}(a)$ and isolate the reward-related terms:

$$\frac{p^{\text{pos}}(a)}{p^{\text{neg}}(a)} \to \frac{\exp(k \cdot v(a)) \cdot p^{\text{pre}}(a)}{\exp(-k \cdot v(a)) \cdot p^{\text{pre}}(a)} = \exp(2k \cdot v(a)) \tag{15}$$

Define the total residue-level preference signals $s^{\mathcal{M}}(a)$ as the natural logarithm of the ratio of positive and negative distributions. Taking the logarithm of both sides of Equation (15) gives:

$$s^{\mathcal{M}}(a) = \log \frac{p^{\text{pos}}(a)}{p^{\text{neg}}(a)} \to 2k \cdot v(a) + C_1 \tag{16}$$

where $C_1$ is a constant absorbing the normalization factors and baseline terms from the proportionality relationship. Letting $\beta' = 2k$, we have:

$$s^{\mathcal{M}}(a) \approx \beta' \cdot v(a) + C_1 \tag{17}$$

Therefore the $s^{\mathcal{M}}(a)$ is a linear approximation of the true reward $v(a)$ when the memory bank is sufficiently large.

*// Step 4: Proving Convergence of $p^{\mathcal{M}}$ to $p^{(\alpha)}$*

**Proposition**: *As $|\mathcal{M}| \to \infty$ and the ratio of positive samples $\rho \to 1$, the practical optimized distribution $p^{\mathcal{M}}$ converges to the theoretical optimal distribution $p^{(\alpha)}$ in total variation (TV) distance.*

*Proof.* From Step 3, we have $s^{\text{total}}(a) = \beta'v(a) + C_1 + \epsilon(a)$, where $\epsilon(a) \to 0$ (the approximation error vanishes as $|\mathcal{M}| \to \infty$ and $\rho \to 1$). Substitute this into the expression for $p^{\mathcal{M}}$:

$$p^{\mathcal{M}}(a_i \mid x_t) \propto p^{\text{pre}} \exp\left(\lambda(\beta'v + C_1 + \epsilon)\right)$$

The constant term $\lambda C_1$ can be absorbed into the proportionality coefficient, so the expression simplifies to:

$$p^{\mathcal{M}}(a_i \mid x_t) \propto p^{\text{pre}} \exp\left(\lambda\beta'v\right) \cdot \exp\left(\lambda\epsilon\right)$$

As $\epsilon \to 0$, we use the Taylor expansion $\exp(\lambda\epsilon) \approx 1 + o(1)$ (where $o(1)$ denotes a higher-order infinitesimal term), leading to:

$$p^{\mathcal{M}}(a_i \mid x_t) \propto p^{\text{pre}} \exp(\lambda\beta'v) \cdot (1 + o(1))$$

The theoretical optimal distribution $p^{(\alpha)}$ from Step 2 is:

$$p^{(\alpha)}(a) \propto p^{\text{pre}}(a) \cdot \exp(\beta \cdot v(a))$$

By setting $\beta = \lambda\beta'$ (aligning the temperature coefficients), the only difference between $p^{\mathcal{M}}$ and $p^{(\alpha)}$ is the higher-order error term $o(1)$, which vanishes as the approximation error converges to zero.

As the error term $o(1)$ vanishes, the TV distance between $p^{\mathcal{M}}$ and $p^{(\alpha)}$ tends to zero:

$$\lim_{\substack{|\mathcal{M}| \to \infty \\ \rho \to 1}} \|p^{\mathcal{M}}(\cdot \mid x_t) - p^{(\alpha)}(\cdot \mid x_t)\|_{\text{TV}} = 0 \tag{18}$$

Therefore the practical distribution converges to the theoretical optimal distribution.

*// Step 5: Proof of Global Sequence Optimality from Single-Step Convergence*

**Proposition**: If $p^{\mathcal{M}}$ converges to $p^{(\alpha)}$ at every diffusion step $t$ and every sequence position $i$, the marginal distribution of the generated sequence $p^{\text{gen}}(x_0)$ converges to the global optimal distribution $p^*$ in TV distance.

*Proof.* Due to the diffusion sampling is a Markov process, which means the next state $x_{t-1}$ depends only on the current state $x_t$ (and not on any earlier states $x_{t+1}, \ldots, x_T$ in the sampling process). This property enables the composition of single-step optimality to approximate the global optimality of the entire sequence generation process.

First, the global optimal distribution $p^*$ is defined as the solution to the test-time reward optimization problem for the entire protein sequence:

$$p^* = \arg\max_p \mathbb{E}_{x \sim p}[r(x)] - \alpha \cdot \text{KL}(p \parallel p^{\text{pre}}) \tag{19}$$

This distribution balances the global sequence reward and the fidelity to the pre-trained distribution for the entire sequence (not just a single position).

Then, by the compositionality of optimal policies in sequential decision processes (a core property of dynamic programming), the convergence of each single-step distribution $p^{\mathcal{M}}$ to $p^{(\alpha)}$ implies the convergence of the marginal distribution of the generated sequence $x_0$:

$$\lim_{\substack{|\mathcal{M}| \to \infty \\ \rho \to 1}} \|p^{\text{gen}}(x_0) - p^*\|_{\text{TV}} = 0 \tag{20}$$

This result holds under standard stability assumptions for diffusion sampling, which ensure that the Markov process converges to the global optimal distribution rather than a local optimum.

The above five steps complete the proof of Theorem 4.1.

# C. Experimental Details

## C.1. Details on Baselines

- **Sequential Monte Carlo (SMC)** (Wu et al., 2023) is a representative single-shot, derivative-free guidance method (Uehara et al., 2025b). It maintains a set of particles during denoising and biases the population toward high-reward regions via importance resampling without requiring gradient information. In SMC, we set $\alpha = 0.05$ because if we choose $\alpha = 0.00$, it just gives a single sample every time step. Refer to Appendix B in (Li et al., 2024).

- **SVDD** (Li et al., 2024) is another single-shot generation method that integrates soft value functions, which look ahead to how intermediate noisy states lead to high rewards in the future, into the standard inference procedure of pre-trained diffusion models. In SVDD, we set $\alpha = 0.0$.

- **Genetic Algorithm (GA)** (Hie et al., 2022) is a basic sequence design approach integrates pre-trained diffusion models into a standard GA pipeline to generate mutated sequences. In GA, we follow the experimental setup of (Uehara et al., 2025a) for this baseline.

- **RERD** (Uehara et al., 2025a) is an iterative refinement framework that performs test-time reward optimization for diffusion models by alternating noising and reward-guided denoising steps to correct errors in protein design. In RERD, we have used parameters $C = 20, B = 10, N = 20$. For the importance sampling step, we have used $\alpha = 0.0$, and for the selection step, we have used $\alpha = 0.2$.

- **MOMST** (ours): To ensure a fair and consistent comparison, we adopt the experimental configurations and hyper-parameter settings following (Uehara et al., 2025a). First, we use EvoDiff (38M) (Alamdari et al., 2023) as the pre-trained model for protein sequence design. We have used parameters: noise level $T = 0.1L$, number of candidate $C = 20$, batch size $B = 10$, iteration number $N = 20$ in multi-objective protein design. For the importance sampling step, we have used $\alpha = 0.0$, and for the selection step, we have used $\alpha = 0.2$.

  In more detail, memory bank size $M_{\text{size}} = 1000$, the threshold for acquiring positive and negative samples is set to $\tau_{\text{high}} : \tau_{\text{low}} = 50 : 50$, and the contrastive scale $\lambda = 0.2$. All experiments were run on a single NVIDIA A100 GPU. More experiment refer to Appendix C.3.

### C.2. Details on Reward Functions

**SS-Match (Secondary Structure Matching).**    This metric quantifies the mean matching probability of all residues between the predicted secondary structure of a protein sequence and the reference secondary structure, where the target structure is represented by a sequence consisting of a ($\alpha$-helices), b ($\beta$-sheets) and c (coils). Herein, we use Biotite (Kunzmann & Hamacher, 2018) to predict the secondary structure (ss) of the generated protein sequences. A score closer to 1 indicates a higher degree of consistency in the secondary structure.

**cRMSD (Coordinate Root-Mean-Square Deviation).**    cRMSD measures the geometric divergence between the generated protein backbone and a reference backbone structure after structural alignment. Typically, $< 2$ Å indicates a highly similar structure. Lower values signify higher structural fidelity, indicating that the model has successfully captured the intended spatial arrangement of the fold.

For **SS-Match** and **cRMSD**, we randomly selected 10 reference proteins from the dataset of (Dauparas et al., 2022) and report the average of the results. Specifically, the ten reference proteins cover native PDB structures (e.g., 1YIU, 5KPH), de novo designed backbones (e.g., HHH_rd1_0033, EHEE_rd1_0101), and computationally evolved intermediates (e.g., EA:run2_0325_0005, XX_run1_0254_0003). This heterogeneous selection ensures that the evaluation of SS-match and cRMSD is performed across a range of topological architectures, encompassing evolutionarily optimized biological folds, idealized synthetic geometric structures, and computationally sampled states.

**Hydrophobicity.**    This metric refers to the degree to which a protein repels water and is primarily defined by the distribution of hydrophobic amino acids within its structure—namely valine, isoleucine, leucine, phenylalanine, methionine, and tryptophan (Chandler, 2002). The optimization of hydrophobicity is achieved by minimizing the average Solvent-Accessible Surface Area (SASA) of the aforementioned hydrophobic residues. Specifically, it drives the correct folding of the protein by directing hydrophobic residues into the protein core, reduces their contact with the surrounding solvent (particularly in polar solvents such as water), and thereby enhances its structural stability and mitigates aggregation.

**Surface Exposure.**    Surface exposure is a critical indicator for measuring the degree to which protein residues are exposed to solvent (usually water) in their three-dimensional structure (Holbrook et al., 1990). Defined based on the contact probability between residue side chains and solvent molecules, it is typically quantified by calculating the Solvent-Accessible Surface Area (SASA) of residues—a higher SASA value indicates a higher degree of residue exposure to the solvent, whereas a lower value means the residue tends to be buried inside the protein. Our goal is to generate proteins that balance water solubility and structural stability by optimizing the exposure state of residues.

**Globularity.**    A globular protein refers to a protein whose three-dimensional structure assumes a compact, nearly spherical state (Pace & Hermans, 1975). It is defined by the spatial arrangement of its backbone atomic coordinates, in which the variance of the distances between these coordinates and the centroid is minimized, thereby forming a highly compact structure. Such proteins are characterized by high structural stability and good water solubility, which distinguish them from fibrous or membrane proteins. Furthermore, this compact conformation helps proteins maintain proper folding and reduce the risk of aggregation.

**pLDDT (Predicted Local Distance Difference Test).** This is a confidence score for evaluating the reliability of local structural conformations in predicted proteins. It is defined by the confidence values assigned to each residue by the model. A higher pLDDT score indicates a higher confidence of the model and a stronger structural stability of the corresponding protein region. In our work, we use ESMFold (Lin et al., 2023) to predict the average pLDDT of the entire sequence.

## C.3. Additional Results

### C.3.1. MORE RESULTS OF MULTI-OBJECTIVE PROTEIN DESIGN

**More Metrics.** We employ the **AvgRank score** per task for comparative analysis, which allows for a more intuitive assessment of each model's overall performance across diverse evaluation metrics (the median of rewards (P50), 95th percentile (P95), and log-likelihood (LL)). The AvgRank score is calculated as follows:

$$\text{AvgRank}_{m,t} = \frac{1}{K_t} \sum_{k=1}^{K_t} R_{m,t,k}$$

**Notations:**

- $\text{AvgRank}_{m,t}$: The average rank score of model $m$ on task $t$

- $K_t$: The number of sub-indicators in task $t$ (e.g., each task in this study includes 3 sub-indicators: P50, P95, and LL)

- $R_{m,t,k}$: The rank of model $m$ on the $k$-th sub-indicator of task $t$ (a rank of 1 indicates the best performance on that indicator)

*Table 7.* AvgRank results for all tasks. A_Rank means AvgRank score. A lower AvgRank score indicates a higher overall performance ranking. Best results are in bold. Here, Tasks 1–4 correspond to the single-objective tasks of hydrophobicity, globularity, ss-match, and cRMSD; Tasks 5–6 correspond to the dual-objective tasks of hydrophobicity & pLDDT and globularity & pLDDT; and Tasks 7–8 correspond to the triple-objective tasks of hydrophobicity & surface exposure & pLDDT and hydrophobicity & globularity & pLDDT.

| MODELS | TASK1 A_RANK↓ | TASK2 A_RANK↓ | TASK3 A_RANK↓ | TASK4 A_RANK↓ | TASK5 A_RANK↓ | TASK6 A_RANK↓ | TASK7 A_RANK↓ | TASK8 A_RANK↓ |
|---|---|---|---|---|---|---|---|---|
| SMC | 3.67 | 3.00 | 4.67 | 3.67 | 4.40 | 3.20 | 3.86 | 3.43 |
| SVDD | 3.67 | 3.00 | 3.00 | 3.67 | 3.20 | 3.00 | 4.00 | 3.57 |
| GA | 4.67 | 5.00 | 4.33 | 5.00 | 3.60 | 5.00 | 2.86 | 3.28 |
| RERD | 1.67 | 2.67 | 1.67 | 2.00 | 2.40 | 2.80 | 3.29 | 3.00 |
| **MOMST** | **1.33** | **1.33** | **1.33** | **1.00** | **1.40** | **1.00** | **1.00** | **1.71** |

**AvgRank Score Results.** As shown in Table 7, we can clearly observe that the proposed MOMST ranks first in AvgRank score across all tasks, which further demonstrates that our method is more adaptable to multi-objective protein design scenarios.

*Table 8.* Diversity results of MOMST.

| MODELS | HYDROPHOBICITY & PLDDT | | | HYDROPHOBICITY & GLOBULARITY & PLDDT | | |
|---|---|---|---|---|---|---|
| | $D_{\text{MAX}}$ (↑) | $D_{\text{MEAN}}$ (↑) | $U@10$ (↑) | $D_{\text{MAX}}$ (↑) | $D_{\text{MEAN}}$ (↑) | $U@10$ (↑) |
| RERD ($T/L = 0.1$) | 0.1867 | 0.1272 | 1 | 0.0267 | 0.0090 | 0.5 |
| MOMST ($T/L = 0.1$) | 0.1667 | 0.1253 | 1 | **0.1667** | **0.1361** | 1 |
| MOMST ($T/L = 0.2$) | **0.4000** | **0.3148** | 1 | **0.3133** | **0.2708** | 1 |

**Task-constrained Diversity Results.** Reward optimization aims to improve protein viability, and it will actively compress into high-fitness regions. Diversity under such strict constraints should be interpreted as task-constrained diversity (Bryant et al., 2021). We measure diversity by Mean Pairwise Normalized Levenshtein Distance $D \in [0, 1]$ and Unique Sequence Rate ($U@10 = (\text{\# unique sequences in top-10})/10$). Under identical editing budgets ($T/L = 0.1$, each sequence allows $0.1L$ editable positions per iteration), MOMST achieves high diversity ($D_{\text{max}} = 0.1667$) and outputs 100% unique sequences ($U@10 = 1$) in the triple-objective task, better than RERD ($D_{\text{max}} = 0.0267, U@10 = 0.5$). When we slightly relax the

mutation budget to $T/L = 0.2$, MoMST's diversity drastically increases ($D_{\max} = 0.4$ in the double-objective task and $D_{\max} = 0.3133$ in the triple-objective task), without losing reward performance (see Table 9). The above results prove that the MoMST maintains meaningful exploration under stringent constraints.

### C.3.2. PARAMETER SENSITIVITY ANALYSIS.

To analyze the impact of key hyperparameters on the performance of MoMST in multi-objective protein design, we conducted a parameter sensitivity analysis as illustrated in Figure 5. The analysis focuses on two critical components: **the positive-to-negative sampling ratio** for residue relative preference signal formation and **the guidance scale** $\lambda$ for probability distribution refinement.

We first evaluate the impact of the sampling ratio used to construct the residue-level relative preference signal, as shown in top row of Figure 5. By fixing the guidance scale at $\lambda = 0.2$ and varying the ratio of trajectories drawn from the successful vs. failed tiers of the Reasoning-aware Memory Bank, we observe that a 50:50 split (positive/negative 50% of the bank, denoted as 50:50) yields the most robust performance across all objectives. As the sampling coverage expands from the extreme tails (positive/negative 5%, denoted as 95:5) toward the median split (50:50), both functional rewards (e.g., globularity or hydrophobicity) and structural confidence (pLDDT) show a consistent upward trend. This suggests that restricting contrastive pairs to only the extreme cases provides insufficient sample diversity, whereas fully utilizing the memory bank provides richer guidance for the model to distinguish between beneficial and detrimental amino acid substitutions.

We further examine the sensitivity of the model to the guidance scale $\lambda$ (Figure 5, bottom row), which modulates the strength of the residue relative preference signal to the pre-trained amino acid probability distribution $p^{pre}$. By fixing the sampling ratio at 50:50 and varying $\lambda$ from 0.2 to 0.8, we find that $\lambda = 0.2$ serves as the optimal threshold. While the guidance signal effectively steers the generation process, excessive weighting (e.g., $\lambda \geq 0.5$) leads to a catastrophic decline in both functional scores and structural integrity. This phenomenon indicates over-optimization, where the guidance signal overpowers the biophysical prior of the pre-trained model, forcing the sequence to drift away from the natural protein manifold.

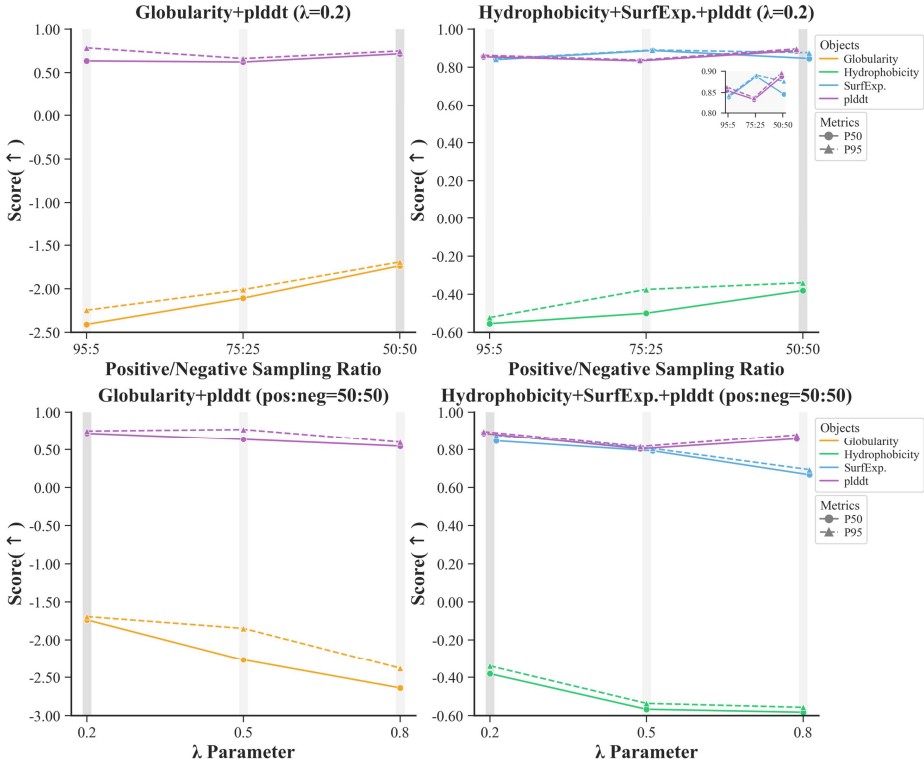

*Figure 5.* Parameter sensitivity analysis.

Additionally, we also experimented with multi-objective design at various noise levels $T$. The MoMST performs well at

*Table 9.* Noisy robustness results of MOMST. Best results are in bold.

| MODELS | HYDROPHOBICITY & PLDDT (↑) | | | | HYDROPHOBICITY & GLOBULARITY & PLDDT (↑) | | | | | |
|---|---|---|---|---|---|---|---|---|---|---|
| | P50 | P95 | P50 | P95 | P50 | P95 | P50 | P95 | P50 | P95 |
| MOMST (T/L=0.04) | -0.5944 | -0.5860 | 0.8730 | 0.8761 | -0.3140 | -0.2830 | **-3.0431** | **-3.0342** | 0.7345 | 0.7545 |
| MOMST (T/L=0.1) | -0.5556 | -0.5406 | **0.9657** | **0.9704** | **-0.2466** | **-0.2270** | -3.2023 | -3.0847 | **0.7657** | 0.7969 |
| MOMST (T/L=0.2) | **-0.4984** | **-0.4777** | 0.8905 | 0.9108 | -0.2885 | -0.2336 | -3.2216 | -3.1676 | 0.7601 | 0.8059 |
| MOMST (T/L=0.5) | -0.5289 | -0.4992 | 0.8756 | 0.9157 | -0.3323 | -0.2944 | -3.5718 | -3.3599 | 0.7219 | **0.8262** |

$T/L = 0.1$ and 0.2 but declines at $T/L = 0.04$ and 0.5, as detailed in Table 9. This result is expected and corroborated by RERD, so this result is robust: a large $T$ reduces the benefits of refinement, while a very small $T$ limits the opportunity for reward-guided decoding. For fair comparison, we followed the experimental setup of RERD and adopted $T/L = 0.1$.

### C.3.3. SENSITIVITY TO MEMORY BANK SCALE

The experimental results in Figure 6 provide empirical evidence for the theoretical convergence established in Theorem 4.1 (refer to Section 4). Specifically, the consistent improvement and subsequent stabilization of **globularity and pLDDT** scores as the bank size $|\mathcal{M}|$ grows from 400 to 1200 align with the theorem's requirements for "sufficient size" and "adequate coverage." This upward trend demonstrates that increasing the memory bank scale effectively drives the guided sampling distribution $p^{\mathcal{M}}$ to converge toward the global reward-optimal solution $p^*$.

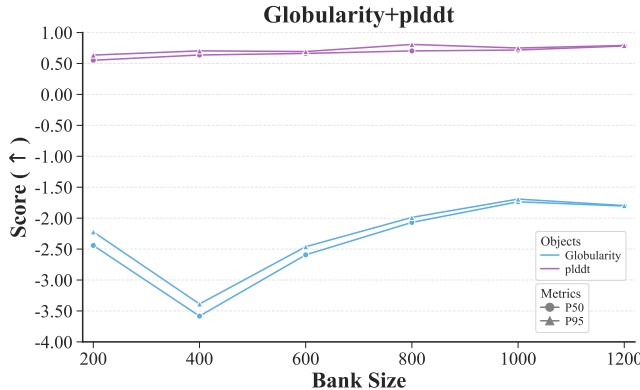

*Figure 6.* Sensitivity analysis of bank size. We take the dual-objective task of globularity & pLDDT as an example and select multiple sets of bank sizes for experiments.

