# OpenReview forum: "Multi-Objective Protein Design via Memory-Aware Test-Time Scaling in Diffusion Models"
_ICML.cc/2026/Conference — ICML 2026 regular_

### Official Review · Reviewer_zwdS · 2026-02-15

**Soundness:** 3
**Presentation:** 3
**Significance:** 3
**Originality:** 3
**Overall Recommendation:** 5
**Confidence:** 3

**Summary:**

This study aims to address three core challenges faced by existing diffusion model-based test-time scaling methods in multi-objective protein design: (1) the inability to learn from iterative history, leading to repeated design errors; (2) over-reliance on successful cases as reward signals, while ignoring critical information provided by failed cases; and (3) difficulty in balancing trade-offs among multiple competing functional objectives. The MOMST framework addresses these issues by introducing a memory bank to extract generalizable reasoning experience from historical iterations, and by leveraging self-contrastive learning to derive rich residue-level relative preference signals from both successful and failed cases to guide protein generation. Furthermore, the framework proposes an inference-time Pareto alignment strategy to resolve objective conflicts, ensuring balance among multiple competing functional objectives. Experiments were conducted on both single-objective and complex multi-objective tasks, and the results demonstrate that MOMST significantly outperforms existing single-shot generation and memory-less iterative optimization baseline methods across all tasks.

**Compliance With Llm Reviewing Policy:**

Affirmed.

**Final Justification:**

My concerns have been resolved. I believe this paper merits acceptance.

**Key Questions For Authors:**

**1. Computational efficiency:** The paper does not evaluate computational overhead. Could the authors provide wall-clock time or FLOPs comparisons between MOMST and baselines (especially RERD) on a single design task?

**2. Pretrained model sensitivity:** All experiments use only EvoDiff. How sensitive is MOMST's performance to the choice of underlying pretrained model? Would results differ significantly with models of different architectures or capabilities?

**3. Comparison with fine-tuning:** MOMST avoids training but introduces a complex test-time pipeline. How does it compare against directly fine-tuning a pretrained model for multi-objective tasks in terms of total compute, development cycle, and deployment flexibility?

**Limitations:**

All experiments were based on computational simulations and metrics. The generated protein sequences lacked wet experimental validation (such as actual expression, folding, and functional assays), therefore their practical biological feasibility and functionality have not yet been definitively proven.

**Strengths And Weaknesses:**

## Strengths:

**Clear problem definition and well-motivated:** The paper clearly identifies three major limitations of current test-time scaling methods in multi-objective protein design and uses these as the entry point to propose the MOMST framework.

**Innovative and systematic methodology:** The proposed solution combines memory mechanisms, self-contrastive learning, and Pareto optimization to form a complete and innovative technical framework. In particular, the design of splitting the memory bank into "elite retention" and "exploration buffer" components effectively balances exploitation and exploration.

**Comprehensive experimental validation:** The paper conducts extensive experiments on single-objective (e.g., hydrophobicity, globularity) and multi-objective (dual-objective, tri-objective) tasks, comparing against multiple types of baselines (SMC, SVDD, GA, RERD), with results consistently demonstrating the superiority of MOMST.

**In-depth ablation studies and analysis:** The paper validates the necessity of each core component—memory bank, self-contrastive learning, and Pareto alignment—through ablation experiments, and further conducts parameter sensitivity analysis and memory bank size impact analysis, strengthening the credibility of the conclusions.

## Weaknesses:

**Method complexity:** The MOMST framework integrates multiple components (memory bank, self-contrastive learning, Pareto optimization), and while it delivers strong performance, this also increases the method's complexity and the number of hyperparameters (e.g., memory bank size, positive/negative sample thresholds, contrastive scale λ, etc.), which may affect its usability and reproducibility.

**Computational overhead:** Compared to single-shot generation methods (e.g., SMC, SVDD), iterative optimization frameworks (including MOMST and RERD) inherently require more inference steps. MOMST additionally maintains and queries the memory bank, performs self-contrastive computations, and conducts Pareto front screening, potentially introducing extra computational and memory overhead. The paper does not discuss efficiency comparisons in detail.

**Dependence on pretrained models:** This method is built upon pretrained diffusion models (e.g., EvoDiff), and its performance ceiling may be constrained by the capabilities of the base model. The paper does not explore the impact of different pretrained models on MOMST's performance.

---

> ### Author Rebuttal · Authors · 2026-03-31
>
> We sincerely appreciate your recognition of our work, especially in clear problem definition, well-motivated, innovative methodology, comprehensive experiments and in-depth analysis. We have provided detailed responses to your concerns below. If our clarifications alleviate your worries, we would be extremely grateful if you could kindly reconsider your evaluation. We also welcome any further discussions on our work.
>
> >**【W2&Q1:Computational efficiency】**
>
> **Response:** MoMST is highly efficient and perfectly comparable to RERD in computational cost. All our experiments strictly align all shared hyperparameters with RERD. **MoMST achieves an almost identical total runtime of ~30 hours** (Complete runtime heatmaps are provided in Figure 1: https://anonymous.4open.science/r/zwdS-0231/README.md). The additional overhead from our new modules is practically negligible.
>
> **Wall-clock time:** All our experiments were run on a single NVIDIA A100 GPU. MoMST processes a batch of 200 sequences (10 proteins × 20 variants) in 240–260 seconds per step. This ~30-hour total runtime shows MoMST’s feasibility for practical, large-scale applications. Detailed calculation and hyperparameters see Table 1:https://anonymous.4open.science/r/zwdS-0231/README.md
>
> We commit to further refining the presentation for greater clarity in the final manuscript, thank you very much!
>
> >**【W1&W3&Q2: Pretrained model and hyperparameters sensitivity】**
>
> **Response:** We sincerely appreciate your suggestions.
>
> **Pretrained model sensitivity:** To ensure fair comparison,  all experiments in the manuscript used the OADM-based EvoDiff (following RERD). To evaluate pretrained model sensitivity, we tested a new base model (DPLM2) on the globularity+plddt task. Results show MoMST outperforms RERD: globularity P50 improves from -2.78 to -2.72, pLDDT P50 from 0.69 to 0.87, indicating significantly higher structural confidence (Table 2:https://anonymous.4open.science/r/zwdS-0231/README.md). This demonstrates the strong architectural robustness of MoMST.
>
> **Hyperparameters Robustness：**MoMST is highly robust. Crucially, all main results across vastly different tasks (Tables 1&2&3) were generated using the exact same configuration ($\tau_{high}:\tau_{low} = 50:50,\lambda = 0.2$**,** Memory=1000). Achieving SOTA without per-task tuning strongly proves its generalizability. Detailed analyses are in Appendix C.3.
>
> **Hyperparameters sensitivity:** Further analysis of hyperparameters (in Appendix C.3).
>
> 1)Contrastive Scale (λ≥0.5): Overpowers the pre-trained biophysical prior, causing generated sequences to drift away from the natural protein manifold (structural collapse).
>
> 2)Memory Size (＜800): Leads to unstable convergence, as established in Theorem 4.1, theoretically, a larger capacity guarantees better convergence.
>
> We will further refine the presentation of these robustness analyses in the final manuscript for greater clarity.
>
> >**【Q3: Comparison between Test-Time Scaling and Fine-Tuning】** **Response:** We appreciate the reviewer highlighting this suggestion. MoMST offers fundamental advantages across all three requested dimensions by operating entirely at test-time:
>
> 1.**Total compute, computational and time efficiency**:
>
> Fine-tuning demands a multi-GPU cluster for memory-intensive backpropagation (e.g., AbNovo[1] uses 8 NVIDIA A100 (80GB) GPUs for training).
>
> In contrast, MoMST relies solely on forward inference, requiring only a single standard A100 GPU, with our experiments showing that exploring 87000 variants takes ~30 hours.
>
> 2.**Development cycle:**
>
> **(1) Data Scarcity**:
>
> Fine-tuning relies on large amounts of paired data that simultaneously satisfy all constraints (e.g., ProteinDT[2]), which is extremely scarce and costly in biology.
>
> MoMST operates in a zero-shot manner, requiring only a black-box scorer.
>
> **(2) Training:**
>
> Fine-tuning requires days of training, and failed objective balancing necessitates restarting the entire pipeline.
>
> MoMST does not require training or parameter updates.
>
> **(3) Evaluation:**
>
> Fine-tuning (Post-hoc Validation) are evaluated after days of training. If objective balancing fails, the entire training pipeline must be restarted.
>
> MoMST evaluates rewards dynamically on-the-fly, actively steering the sequence trajectory in real-time.
>
> 3.**Deployment flexibility**:
>
> Fine-tuning requires retraining the model for any new combination of multi-objective constraints (e.g., AbNovo[1] needs to adjust the constrained preference optimization loss).
>
> In contrast, MoMST can flexibly apply any reward for attribute optimization of proteins and dynamically balance new objectives without updating model parameters (e.g., weights for hydrophobicity, globularity, pLDDT can be set to 1,1,1 or 1,2,3 during design).
>
> [1]Ren M. et al. Multi-objective antibody design with constrained preference optimization. ICLR2025.
>
> [2]Liu, S. et al. A text-guided protein design framework[J]. Nature Machine Intelligence, 2025, 7(4): 580-591.

---

> > ### Author Rebuttal · Reviewer_zwdS · 2026-04-01
> >
> > My concerns have been resolved. I believe this paper merits acceptance.

---

> > > ### Author Response · Authors · 2026-04-02
> > >
> > > We are pleased to have addressed your concerns. We sincerely appreciate your time, your insightful feedback, as well as your support and recommendation for acceptance.

---

### Official Review · Reviewer_wocH · 2026-03-12

**Soundness:** 3
**Presentation:** 3
**Significance:** 3
**Originality:** 3
**Overall Recommendation:** 5
**Confidence:** 4

**Summary:**

This paper presents MOMST, a training-free framework for multi-objective protein design that operates through memory-aware self-contrastive learning with test-time scaling in diffusion models. Building upon prior reward-guided diffusion methods (notably RERD), MOMST introduces three contributions: a memory bank that stores generalizable experience from historical iterations to prevent repetitive design errors, a self-contrastive learning mechanism that extracts residue-level preference signals from both successful and failed trajectories, and an inference-time Pareto alignment strategy that balances competing objectives. Evaluations on single-objective and multi-objective protein design tasks demonstrate strong performance across multiple baselines, supported by theoretical analysis and ablation studies.

**Compliance With Llm Reviewing Policy:**

Affirmed.

**Final Justification:**

Thanks for the feedback. I will raise my score.

**Key Questions For Authors:**

1. Computational fairness: Can you provide wall-clock time and/or FLOPs comparisons for all methods in Tables 1, 2, and 3? Were all baseline methods given the same computational budget (e.g., same number of diffusion sampling steps, same number of candidate evaluations)? If not, how do results change when budgets are equalized?

2. Memory bank ablation: Can you provide a dedicated ablation study on the two-tier memory bank architecture? Specifically, what is the individual contribution of the "recency tier" versus the "quality tier"? Why is the 50% split optimal—was this tuned, and how sensitive is performance to this hyperparameter (e.g., 30/70 or 70/30 splits)?

3. Pareto alignment exposition: Can you expand the discussion of the inference-time Pareto alignment strategy in the main text, including its formal definition, algorithmic details, and a comparison with alternative multi-objective balancing strategies ?

4. Scaling behavior analysis: Can you include experiments that characterize performance as a function of computational budget (e.g. candidate evaluations) for each task? Is there a point of diminishing returns, and can you provide practical budget recommendations for end users?

**Limitations:**

Yes

**Strengths And Weaknesses:**

**Strength**

MOMST offers a practical and elegantly simple set of innovations layered atop the existing RERD framework, combining memory-aware modeling, preference learning from both positive and negative examples, and multi-objective optimization into a cohesive test-time scaling paradigm. The idea of maintaining a memory bank to break the cycle of repetitive design errors is well-motivated and addresses a genuine shortcoming of current memoryless diffusion guidance methods. The self-contrastive learning component is particularly appealing because it leverages failed trajectories—typically discarded information—to construct informative residue-level preference signals, which is both data-efficient and biologically meaningful. The paper is further strengthened by its theoretical analysis, which provides formal grounding for the proposed approach, and by a comprehensive experimental setup that includes a decent number of baselines and ablation studies across both single-objective and multi-objective protein design benchmarks. The training-free nature of the approach makes it highly flexible for practitioners who need to rapidly reconfigure objective combinations without incurring retraining costs.


**Weakness**

Despite its merits, the paper has several notable weaknesses that limit the strength of its claims. First, given that the paper is fundamentally about test-time scaling of diffusion models, the absence of a fair computational cost comparison across all baselines in Tables 1, 2, and 3 is a significant oversight—it is unclear whether baseline methods were allocated equivalent computational budgets, making performance comparisons potentially misleading. Second, the two-tier memory bank system, particularly the "recency tier" and the specific choice of 50% as the partition threshold, lacks sufficient ablation analysis and justification; without this, the design appears somewhat arbitrary. Third, the inference-time Pareto alignment strategy—arguably one of the paper's three core contributions—is not introduced or discussed with adequate depth in the main text, which undermines the reader's ability to assess its novelty and effectiveness. Finally, since the work positions itself squarely within the test-time scaling paradigm, there is a conspicuous lack of experimentation characterizing the relationship between computational budget and expected performance improvement across tasks; without such analysis or practical recommendations for users regarding computational budget allocation, the paper's practical guidance for adoption remains incomplete.

---

> ### Author Rebuttal · Authors · 2026-03-31
>
> We greatly appreciate the reviewer’s recognition of our work and constructive comments. We provide detailed responses to the concerns below.
>
> >**【W1&Q1:Computational fairness】**
>
> **Response: We carefully ensured computational fairness, using identical values for all shared hyperparameters (including 20 candidate evaluations), strictly following RERD.** Regarding diffusion sampling steps, MoMST shares the exact same configuration as its diffusion-based counterparts (e.g., RERD); deviations in other baselines inherently stem from their fundamentally different generation paradigms (e.g., one-shot vs. iterative methods) rather than unfair tuning.
>
> **Sampling step:** For fair comparison, we standardized settings (L=150, 20 candidates, 20 iterations, because baseline plateau beyond this, see Fig. 4). The total sampling steps per sequence in varying paradigms are: GA (285), One-shot methods (SMC/SVDD: 150), and iterative methods (RERD/**MoMST(ours)**: 435). Crucially, MoMST uses the exact same sampling budget as RERD but achieves superior multi-objective balance. Detailed calculation and hyperparameters see Table 1: https://anonymous.4open.science/r/wocH-07A1/README.md
>
> **Wall-clock time:** All our experiments were run on a single NVIDIA A100 GPU. MoMST is highly efficient and comparable to RERD, it processes a batch of 200 sequences (10 proteins × 20 variants) in 240–260 seconds per step. The entire experiment was completed in ~30 hours, demonstrating MoMST’s strong feasibility for practical, large-scale applications. Complete runtime heatmaps are provided in Figure 1: https://anonymous.4open.science/r/wocH-07A1/README.md
>
> >**【W2&Q2:Memory bank ablation & 50% split】**
>
> **Response:** Experiment results show that 50/50 split is optimal. Both tiers are essential: the "quality tier" stores positive samples to guide the positive optimization direction, while the "recency tier" stores negative samples to provide hard negative boundaries (removing the negative tier entirely causes a severe performance drop, see Table 4 "w/o self-contrastive").
>
> 1. Sensitivity to 70/30 and 30/70 splits ablation: We evaluated new memory splits (Table 2:https://anonymous.4open.science/r/wocH-07A1/README.md), The 50/50 split achieves the best overall multi-objective balance compared to 70/30 or 30/70. In the later stages of optimization, the information derived from purely extreme positive/negative samples is limited, the 50/50 split fully leverages the "middle-boundary" samples, which are more effective in capturing the Pareto boundary comprehensively.
>
> 2. Sampling Size Ablation (Appendix Fig. 5): Our existing Appendix Figure 5 further supports this. We tested taking equal positive/negative samples at 5% (95:5), 25% (75:25), and 50% (50:50) capacities. Performance strictly improves as the capacity increases to 50%. Using only extreme top/bottom samples (e.g., 5%) provides limited guidance.
>
> >**【W3&Q3:Pareto alignment exposition】**
>
> **Response:** We will revise Section 3.5 and add explanations in the algorithm section to improve readability:
>
> 1.Formal definition: A Pareto-guided robust sampling strategy that prioritizes non-dominated sequences by their worst-performing objective to promote balanced multi-objective optimization.
>
> 2.Algorithmic details:
>
> (1) Rank-normalize all objective scores within the batch.
>
> (2) Extract the Pareto set (non-dominated sequences) to retain compromise-optimal solutions.
>
> (3) Merge the current set with historical elites to prevent quality regression.
>
> (4) Resample from this global Pareto pool using the reward on the worst-ranked objective as the robustness score and apply partial remasking to start the next refinement round.
>
> 3.Comparison with alternative strategies: Unlike the weighted-sum approach (e.g., RERD), which often suffers from "reward domination" (where one extremely high score masks failures in other metrics), our Pareto strategy explicitly pulls up the weakest objective. This strictly prevents biased sequences without requiring manual weight tuning.
>
> >**【W4&Q4:Scaling behavior analysis】**
>
> **Response**: We sincerely thank the reviewer for this highly practical question.
>
> 1.Scaling with Iterations: We have implicitly provided this analysis in Figure 4, which plots rewards against iteration numbers. MoMST exhibits sharp early gains. A clear point of diminishing returns occurs around iteration 15, where the performance curve plateaus.
>
> 2.Scaling with Candidate: To directly address the reviewer's suggestion, we conducted an additional ablation on the number of candidates generated per step ($C=10,20,30,40$), summarized in the Table 3:https://anonymous.4open.science/r/wocH-07A1/README.md . When candidates=20, it is the point of optimal profit.
>
> 3.Practical Recommendations: Based on this scaling, MoMST is strong robustness and does not require prohibitive computational resources. For reliable use, we recommend the comprehensive hyperparameters detailed in Table 1.

---

> > ### Author Rebuttal · Reviewer_wocH · 2026-04-01
> >
> > Thanks for the feedback. I will raise my score.

---

> > > ### Author Response · Authors · 2026-04-03
> > >
> > > We are pleased to have addressed your concerns and sincerely appreciate your insightful feedback, as well as your willingness to raise the score in support of our manuscript.
> > >
> > > Thank you again for your time. We will ensure all of your suggestions are fully incorporated into the final version.

---

### Official Review · Reviewer_3uQG · 2026-03-13

**Soundness:** 3
**Presentation:** 2
**Significance:** 3
**Originality:** 2
**Overall Recommendation:** 4
**Confidence:** 4

**Summary:**

This paper studies multi-objective protein sequence design under black-box structural/property rewards, focusing on test-time scaling of a pre-trained diffusion model so that new and changing objectives can be handled without retraining. The authors propose MOMST, which augments iterative test-time reward optimization with (i) a reasoning-aware memory bank that stores both elite high-reward samples and recent exploration samples, (ii) self-contrastive learning that extracts residue-level relative preference signals by contrasting successful and failed candidates, and (iii) an inference-time Pareto alignment strategy to maintain balanced trade-offs among multiple objectives. Experiments on single-objective and multi-objective protein design tasks show consistent improvements over single-shot guidance and memoryless iterative baselines, and the paper further reports additional metrics (e.g., fold integrity and sequence plausibility) as supplementary validation.

**Compliance With Llm Reviewing Policy:**

Affirmed.

**Final Justification:**

My initial concerns are properly addressed by the authors during the rebuttal period, so I keep my positive score.

**Key Questions For Authors:**

1. What is the total number of reward evaluations (and any structure prediction calls, if applicable) per designed sequence for MOMST vs. each baseline? Can you report wall-clock time or normalized compute?
2. How sensitive are the results to the main hyperparameters (memory size, elite/recency split, positive/negative thresholds, contrastive scale, and Pareto alignment settings)? Are there regimes where performance collapses or becomes unstable?
3. How do you ensure that “failed/low-reward” samples provide reliable residue-level negative signals rather than simply reflecting noise from early iterations? Do you use any filtering to avoid misleading negatives?
4. Does the method preserve sequence/structure diversity across the Pareto set, or does the memory mechanism risk mode collapse toward a narrow region? Can you include diversity metrics or qualitative analysis?

**Limitations:**

Yes

**Strengths And Weaknesses:**

# Strengths:
- The proposed framework is conceptually coherent: it explicitly targets key pain points in test-time reward optimization for diffusion-based protein design—leveraging interaction history, incorporating failure signals, and addressing multi-objective trade-offs—rather than only changing one component.
- The method is supported by ablations that remove each main component (memory bank / self-contrastive learning / Pareto alignment), showing noticeable drops in multi-objective balance and/or structural reliability when components are removed.
- The experimental protocol includes both single- and multi-objective settings, plus additional evaluation metrics beyond the core rewards (helpful for assessing foldability / plausibility rather than only “reward hacking”).

# Weaknesses:
- The system introduces multiple moving parts (memory bank construction, contrastive thresholds, Pareto alignment strategy), and it is not fully clear how robust performance is across a wider range of hyperparameter choices, tasks, or reward definitions (even if some sensitivity analyses are provided).
- Computational cost is potentially high (iterative sampling + repeated reward evaluations; possibly structure prediction calls depending on the metric). A clearer accounting of #reward calls / wall-clock / compute vs. baselines would strengthen the soundness of the empirical comparison.
- There are noticeable writing/formatting issues (e.g., punctuation around equations, some awkward phrasing). These are fixable but currently reduce readability.

---

> ### Author Rebuttal · Authors · 2026-03-30
>
> We greatly appreciate the reviewer’s recognition of our work and constructive comments. We provide detailed responses to the concerns below.
> ***
> >**【W1&Q2:Hyperparameters sensitivity and performance robustness】**
>
> **Response:**
>
> **Robustness**:MoMST is adequately robust. Crucially, all main results across vastly different tasks (Tables 1-3) shared the exact same configuration (memory size=1000,split/thresholds:$\tau_{high}:\tau_{low} = 50:50$,contrastive scale:$\lambda = 0.2$). Achieving SOTA without per-task tuning strongly proves its generalizability. Detailed analyses are in Appendix C.3.
>
> **Sensitivity:**
>
> 1)Contrastive Scale (λ≥0.5): Overpowers the pre-trained biophysical prior, causing generated sequences to drift away from the natural protein manifold (structural collapse).
>
> 2)Memory Size (＜800): Leads to unstable convergence, as established in Theorem 4.1, theoretically, a larger capacity guarantees better convergence.
>
> We will further refine the presentation of these robustness analyses in the final manuscript for greater clarity.
>
> >**【W2&Q1:Computational cost and reward evaluations】**
>
> **Response:**
>
> **Reward evaluations:** For fair comparison, we standardized settings (L=150, 20 candidates, 20 iterations, because baseline plateau beyond this, see Fig. 4). The total reward evaluations per sequence are: GA (20), One-shot models (SMC/SVDD: 3000), and Iterative models (RERD/**MoMST**: 8700). Crucially, MoMST uses the exact same evaluation budget as RERD but achieves far superior multi-objective balance. Detailed calculation Table 1: https://anonymous.4open.science/r/3uQG-47CB/README.md
>
> **Wall-clock time:** All our experiments were run on a single NVIDIA A100 GPU. MoMST is highly efficient, it processes a batch of 200 sequences (10 proteins × 20 variants) in 240–260 seconds per step. The entire experiment was completed in ~30 hours, demonstrating MoMST’s strong feasibility for practical, large-scale applications. Complete runtime heatmaps are provided in https://anonymous.4open.science/r/3uQG-47CB/README.md
>
> >**【W3:Writing and Formatting】**
>
> **Response:** We thank the reviewer for the constructive feedback. We will address these concerns in the final version for better readability.
>
> >**【Q3:Reliability of Negative Signals from Failed Samples】**
>
> **Response:** To ensure negatives provide reliable boundary signals rather than early noise, we employ mechanisms as follows:
>
> 1.**Sliding Window Elimination:** The Exploration Buffer of memory bank employs a sliding-window mechanism to store the most recent low-scoring samples. Consequently, early-stage, low-quality noise is naturally purged over time.
>
> 2.**Dynamic Definition of Negatives:** We define failures by sampling positives and negatives based on relative percentiles of current memory capacity, not fixed thresholds. As sequence quality improves, the negative bar tightens, ensuring later "failures" are hard negatives near the Pareto boundary.
>
> 3.**De-duplication and Continuous Re-sorting:** We delete identical sequences and re-rank the rest at each step to align the memory bank with the latest exploration boundaries.
>
> As shown in our ablation study (Table 4), removing this negative guidance severely drops performance, proving these filtered signals are reliable and essential.
>
> >**【Q4:Diversity analysis】**
>
> **Response:** Thanks for this thoughtful comment. We measure diversity by Mean Pairwise Normalized Levenshtein Distance$D \in[0,1]$(=1-pairwise sequence identity) and Unique Sequence Rate ($U_{@10}$ ). As in Table 2:https://anonymous.4open.science/r/3uQG-47CB/README.md, **MoMST effectively avoids model collapse ($D_{max}=0.4, U_{@10}=1$) and maintains meaningful exploration under stringent constraints.**
>
> 1.Analysis: Under identical editing budgets (T/L=0.1, editable sequence =150×0.1=15 positions/iteration), MoMST achieves high diversity ($D_{max}=0.1667$) and outputs 100% unique sequences ($U_{@10}=1$), better than RERD ($D_{max}=0.0267,U_{@10}=0.5$). When we slightly relax the mutation budget to T/L=0.2, MoMST's diversity drastically increases ($D_{max}=0.4$, ~60 AA differences), without losing reward performance (refer to【W3&Q6】of reviewer 9ZUb).
>
> 2.Task-constrained diversity: Multi-objective constraints naturally shrink the viable sequence space. **Reward optimization aims to improve protein viability, and it will actively compress into high-fitness regions. Diversity under such strict constraints should be interpreted as "task-constrained diversity", not unconditional diversity.** As shown in [1], local editable tasks should prioritize evaluating the ability to explore local subspaces rather than overall diversity, focusing on positions 561–588 segment, to assess controllable diversity. Since MoMST mutates at random positions, we evaluate its full-length diversity, which still yields excellent diversity.
>
> [1]Bryant D H. et al. Deep diversification of an AAV capsid protein by machine learning[J]. Nature Biotechnology, 2021, 39(6): 691-696.

---

> > ### Author Rebuttal · Reviewer_3uQG · 2026-04-03
> >
> > I thank the authors for their clear and detailed rebuttal. I also read other reviewers' comments and the authors' replies. Overall, the rebuttal satisfactorily addresses many of my initial concerns and strengthens my confidence in the technical validity of the approach. I would like to keep my score to support the acceptance of this work.

---

> > > ### Author Response · Authors · 2026-04-04
> > >
> > > We greatly appreciate your careful and thorough review, as well as your thoughtful attention to the comments from other reviewers and our corresponding responses. We are pleased that our rebuttal has addressed your concerns.
> > >
> > > Thank you for your time and for maintaining your positive score. We will carefully incorporate your valuable suggestions into the final manuscript.

---

### Official Review · Reviewer_9ZUb · 2026-03-16

**Soundness:** 3
**Presentation:** 3
**Significance:** 2
**Originality:** 2
**Overall Recommendation:** 3
**Confidence:** 3

**Summary:**

The authors propose to generate samples optimizing for several scalar metrics from diffusion models. To do so, they resort to a buffer split into two components: an elite buffer (best generations according to the optimized metrics) and an exploration buffer (latest generations). The sampling is steered towards high-performing sequences through a contrastive mechanism that alters the sampling probability of the pretrained diffusion model.

**Compliance With Llm Reviewing Policy:**

Affirmed.

**Key Questions For Authors:**

Q1. What is M+ (line 289)? Is this equivalent to Mhigh defined in Equation 5?

Q2. Can the proposed method be used for efficient conditioning beyond optimization?

Q3. Can the authors comment on the impact of the chosen metrics on the optimization process? How sensitive is performance to metric choice?

Q4. In Table 4, does "Inference-Time Pareto Alignment" refer to the treatment of the elite buffer described in Section 3.5?

Q5. Can the authors report on the diversity of obtained protein sequences? For instance, average pairwise sequence identity or clustering statistics across the Pareto set.

Q6. Most works studying sampling strategies for biological sequence optimization consider fitness landscapes based on noisy experimental data. Can the authors comment on the robustness of their method to noisy reward signals, relevant for practical protein redesign applications?

**Limitations:**

No.

**Strengths And Weaknesses:**

Strengths

*Clarity*
The paper is clear and straightforward to follow.

*Theoretical Motivation*
 The buffer intuition is mathematically motivated by showing that the final distribution under infinite buffer size and high quality converges towards the distribution of sequences maximizing the reward.

*Empirical Validation*
 The experiments show that this type of inference-time scaling performs well compared to baselines on the investigated optimization metrics.

Weaknesses

**Limited Novelty in Context of Evolutionary Methods**
 While the idea of using a buffer is relatively new for inference of diffusion models, it is a well-known technique in population/repertoire-based genetic algorithms. See all MAP-Elites-based methods and in particular "Quality diversity optimization for one-shot biological sequence design," where the sampler can be any diffusion model. The relationship to this body of work should be discussed.

**Missing Diversity Analysis:**
 The paper does not report diversity metrics for the generated sequences, making it difficult to assess whether the optimization collapses to similar solutions or maintains meaningful exploration.

**Robustness to Noise Not Addressed:**
 Most works studying sampling strategies for biological sequence optimization consider fitness landscapes based on noisy experimental data. While the authors' choice of metrics adequately demonstrates efficiency, the robustness to noise for practical protein redesign applications is not discussed.

---

> ### Author Rebuttal · Authors · 2026-03-30
>
> We sincerely appreciate your recognition of our work, especially in theoretical motivation, empirical validation and clear presentation! We have provided detailed responses to your concerns below. If our clarifications alleviate your worries, we would be extremely grateful if you could kindly reconsider your evaluation. We also welcome any further discussions on our work！
> ***
> >**【W1:Clarification on novelty in context of evolutionary methods】**
>
> **Response:** Thank you for highlighting "Quality diversity optimization...". We will gladly cite and discuss it. While population-based algorithms use buffers, MoMST fundamentally differs in three key ways:
>
> **1.Storage (Elites vs. Elites + Failures):** MAP-Elites passively archives only successful elites. MoMST maintains a dynamic dual-tier bank storing both high-reward elites and recent failures (hard negatives) to define functional boundaries.
>
> **2.Usage (Black-box mutation vs. Conscious mutation):** MAP-Elites samples parents for 'black-box mutation'. MoMST distills preference signal from historical trajectories (successful and failed samples) to enable 'conscious mutation'.
>
> **3.Generative** **Paradigm** **(One-shot vs. Reasoning-aware iterative):** Unlike one-shot models in "Quality Diversity Optimization for One-Shot Biological Sequence Design," which lack sequential error correction, MoMST integrates memory into reverse diffusion for iterative refinement and repetitive error prevention.
>
> >**【W2&Q5:Missing Diversity Analysis】**
>
> **Response:** Thanks for this thoughtful comment. We measure diversity by Mean Pairwise Normalized Levenshtein Distance$D \in[0,1]$(=1-pairwise sequence identity) and Unique Sequence Rate ($U_{@10}$ ). As in Table 1:https://anonymous.4open.science/r/9ZUb-4D8A/README.md, **MoMST effectively avoids model collapse ($D_{max}=0.4, U_{@10}=1$) and maintains meaningful exploration under stringent constraints.**
>
> 1.Analysis: Under identical editing budgets (T/L=0.1, editable sequence =150×0.1=15 positions/iteration), MoMST achieves high diversity ($D_{max}=0.1667$) and outputs 100% unique sequences ($U_{@10}=1$), better than RERD ($D_{max}=0.0267,U_{@10}=0.5$). When we slightly relax the mutation budget to T/L=0.2, MoMST's diversity drastically increases ($D_{max}=0.4$, ~60AA differences), without losing reward performance (refer to【W3&Q6】).
>
> 2.Task-constrained diversity: Multi-objective constraints naturally shrink the viable sequence space. **Reward optimization aims to improve protein viability, and it will actively compress into high-fitness regions. Diversity under such strict constraints should be interpreted as "task-constrained diversity", not unconditional diversity.** As shown in [1], local editable tasks should prioritize evaluating the ability to explore local subspaces rather than overall diversity, focusing on positions 561–588 segment, to assess controllable diversity. Since MoMST mutates at random positions, we evaluate its full-length diversity, which still yields excellent diversity.
>
> >**【W3&Q6:Noisy Robustness】**
>
> **Response:**  Thanks for your insightful suggestions. We experimented with multi-objective design at various noise levels T. **The MoMST performs well at T/L=0.1 and 0.2** but declines at 0.04 and 0.5 (e.g., editable sequence length=150×0.1=15 positions/iteration), details in Table 2: https://anonymous.4open.science/r/9ZUb-4D8A/README.md. **This result is expected and corroborated by RERD, so this result is** **robust****: a large T reduces the benefits of refinement, while a very small T limits the opportunity for reward-guided decoding. For fair comparison, we followed RERD's setup using T/L=0.1.**
>
> >**【Q1:For $M^+$】**
>
> **Response:** Yes, $M^+$ equivalent to $M_{high}$. We will unify the notation in the final manuscript.
>
> >**【Q2:For other conditioning】**
>
> **Response:** Yes. By formulating any target condition as a reward function, MoMST naturally extends to efficient conditional generation.
>
> >**【Q3:The impact of the chosen metrics】**
>
> **Response:** The performance of MOMST is not sensitive (fairly robust) to the choice of metrics. We achieve this by applying non-parametric **Rank** **Normalization** to all raw rewards and extracting contrastive signals using **percentile thresholds** ($M_{high}$ and $ M_{low}$)  rather than absolute reward differences. This ensures stable relative evolutionary pressure, preventing any single metric's magnitude from dominating the multi-objective optimization.
>
> >**【Q4:Inference-Time Pareto Alignment of Table 4】**
>
> **Response:** Yes, the "Inference-Time Pareto Alignment" in Table 4 refers to Section 3.5. It encompasses Pareto front filtering, merging non-dominated sets with historical elites, and Pareto sampling. As Table 4 shows, removing it severely degrades multi-objective balance. We will clarify this mapping in the revised Ablation Study.
>
> [1]Bryant D H. et al. Deep diversification of an AAV capsid protein by machine learning[J]. Nature Biotechnology, 2021, 39(6): 691-696.

---

> > ### Author Rebuttal · Reviewer_9ZUb · 2026-04-04
> >
> > I acknowledge the author's rebuttal and thank them for their thorough review.
> >
> > regarding the diversity: if i read things correctly a diversity of 15% as defined by your rebuttal means a pairwise similarity of 85% - which i believe says that the model only sample homologous proteins.
> > Also, in my opinion without clear positioning to population based methods, i find it  difficult to assess the novelty of the method or its relevance.

---

> > > ### Author Response · Authors · 2026-04-04
> > >
> > > Thank you for your continued engagement. We would like to quickly clarify two critical points:
> > > ***
> > >
> > > **1.Regarding Diversity:** Thank you for the clarification. We agree that our previous wording on diversity was too strong, and we should have framed the result more carefully. In our setting, diversity is measured on the full-length 150-aa sequences, while each iteration performs local editing under a strict mutation budget (T/L=0.1), multi-objective reward optimization, and elite selection. Therefore, this quantity should be interpreted as **"task-constrained full-length diversity"**, rather than compared directly to unconstrained de novo generation. In this iterative constrained-optimization regime, moderate full-length similarity among final elites is not unexpected and does not by itself indicate collapse. Importantly, when the mutation budget is slightly relaxed from T/L=0.1 to T/L=0.2, the pairwise full-length diversity increases substantially to ~40% (about 60 amino-acid differences under our metric), while optimization performance is maintained (e.g. hydrophobic P95 is -0.4777, higher than -0.5406 (T/L=0.1) in hydrophobic+plddt task). This suggests that MoMST retains meaningful exploratory capacity under stricter budgets, rather than degenerating into trivial near-duplicates. We will revise the manuscript to describe this more precisely, and we thank the reviewer for bringing this to our attention.
> > >
> > > **2.Regarding Positioning:** We also thank the reviewer for this insightful comparison to population-based / QD methods such as MAP-Elites, which helps sharpen the framing of our method. MoMST was originally motivated less by classical evolutionary algorithms than by memory-style mechanisms in modern AI systems: a single diffusion model that can effectively “remember” past high-scoring explorations (elites) and learn from failures (hard negatives) to improve subsequent inference. However, our claim is not that buffers or archives are new per se, nor that MoMST is unrelated to population-based methods. Rather, the key novelty is where this memory enters the algorithm: archive-derived preference signals from both successful and failed trajectories are injected directly into the inference-time reverse diffusion process, so that historical feedback shapes the denoising trajectory itself and enables a form of inference-time self-correction. In contrast, classical population-based methods typically operate on completed sequences through external selection, mutation, and replacement. We will revise the paper to make this framing more explicit, positioning MoMST as an archive-guided diffusion refinement method with clear conceptual connections to population-based / QD methods, but a distinct mechanism of generation-time control.
> > >
> > > Thank you once again!

---

### Decision · Program_Chairs · 2026-04-30

**Decision:**

Accept (regular)

**Comment:**

This paper proposes a framework for multi-objective protein design via test-time scaling of diffusion models. The reviewers highlighted the coherent integration of memory, self-contrastive learning from successes and failures, and inference-time Pareto alignment, as well as strong empirical results supported by ablations. The training-free, inference-time nature of the method was viewed as particularly attractive.

One reviewer raised concerns regarding the interpretation of diversity results and the positioning relative to population-based and quality-diversity methods. In rebuttal and follow-up discussion, the authors clarified that diversity should be interpreted as task-constrained full-length diversity, provided additional evidence showing increased diversity under relaxed mutation budgets, and committed to narrowing their claims accordingly. They also clarified that the core novelty lies in injecting archive-derived preference signals (including failures) directly into the reverse diffusion process, rather than in post-hoc selection as in population-based methods. These clarifications satisfied the other reviewers, who maintained or strengthened their positive recommendations.

Overall, the remaining issues are primarily about framing and exposition rather than soundness or significance. With the proposed revisions, the paper meets ICML acceptance criteria.